# Inherited and Acquired Rhythm Disturbances in Sick Sinus Syndrome, Brugada Syndrome, and Atrial Fibrillation: Lessons from Preclinical Modeling

**DOI:** 10.3390/cells10113175

**Published:** 2021-11-15

**Authors:** Laura Iop, Sabino Iliceto, Giovanni Civieri, Francesco Tona

**Affiliations:** Department of Cardiac Thoracic Vascular Sciences and Public Health, University of Padua, Via Giustiniani, 2, I-35124 Padua, Italy; sabino.iliceto@unipd.it (S.I.); giovanni.civieri@gmail.com (G.C.)

**Keywords:** cardiac conduction system, rhythm disturbances, preclinical modeling, in vivo, in vitro, in silico

## Abstract

Rhythm disturbances are life-threatening cardiovascular diseases, accounting for many deaths annually worldwide. Abnormal electrical activity might arise in a structurally normal heart in response to specific triggers or as a consequence of cardiac tissue alterations, in both cases with catastrophic consequences on heart global functioning. Preclinical modeling by recapitulating human pathophysiology of rhythm disturbances is fundamental to increase the comprehension of these diseases and propose effective strategies for their prevention, diagnosis, and clinical management. In silico, in vivo, and in vitro models found variable application to dissect many congenital and acquired rhythm disturbances. In the copious list of rhythm disturbances, diseases of the conduction system, as sick sinus syndrome, Brugada syndrome, and atrial fibrillation, have found extensive preclinical modeling. In addition, the electrical remodeling as a result of other cardiovascular diseases has also been investigated in models of hypertrophic cardiomyopathy, cardiac fibrosis, as well as arrhythmias induced by other non-cardiac pathologies, stress, and drug cardiotoxicity. This review aims to offer a critical overview on the effective ability of in silico bioinformatic tools, in vivo animal studies, in vitro models to provide insights on human heart rhythm pathophysiology in case of sick sinus syndrome, Brugada syndrome, and atrial fibrillation and advance their safe and successful translation into the cardiology arena.

## 1. Introduction

Preclinical modeling represents an essential path towards the understanding of the complex molecular modifications, which result in a disease phenotype. It is also fundamental for the design of more effective diagnostic tools and therapeutic strategies to be applied in clinical practice. The recapitulation of human pathophysiology by means of in vitro, in vivo and in silico models has strongly contributed to the study of the pathogenesis of many diseases in the cardiovascular field and in particular of the heart rhythm, which still remain among the primary causes of death worldwide [1]. Heart rhythm disorders are a very heterogeneous group of disturbances that might manifest already in the fetal life, as caused by congenital mutations, or acquired in the adult time, following external triggers or concurrently with other comorbidities [2,3,4]. In this review, the strategies so far applied for the modeling of heart rhythm diseases will be discussed with the aim to evaluate their ability to recapitulate typical pathognomonic signs.

## 2. The Embryological Development, Anatomy, and Physiology of the Heart Rhythm

The anatomical discovery of the sinoatrial node (SAN), atrioventricular node (AVN), His bundle, and Purkinje fibers dates back to the end of the nineteenth century [5,6,7,8]. However, the research interest in the cardiac conduction system (CCS) resumed and flourished during the 1960s [9,10]. With the progressive availability of new investigative tools and technologies, several embryological studies were dedicated to the comprehension in which cells contribute to its formation, which molecular pathways are regulated during its developmental phases, and how the conduction system varies during the fetal life.

A first answer to these questions was offered by the seminal histological work by Viragh and Challice, who carefully analyzed the multiple components of the conduction system in the developing mouse and avian embryos. Primordial signs of SAN and AVN formation in the mouse were observed already at embryonic day E8 with distinct cell contributions for the two nodes. Upon this initial, but unconcluded nodal organization, heart beating starts (E9). While the primitive cardiomyocytes deputed to the pacing function continue to organize and acquire a differentiated phenotype in the central portions of the nodes (E10–E16), connective fibrous tissue begins to create a capsule-like structure around them and contribute to the electrical isolation of atria and ventricles (E12) [11,12,13,14,15].

Lineage tracing studies in mice and other mammals allowed the clarification of the origin of the CCS cell contributors, identify the hierarchical transcriptional programs governing them, follow their fate more precisely, and specify their spatiotemporal evolution during embryonic development. Several transgenic models were generated to monitor the cells in which a transcription factor pivotal for the CCS development is activated during embryogenesis and post-natal life, the contribution of specific stem cells or progenitors from cardiac and extra-cardiac sources, as well as the structural and functional connections established between the CCS and the adjacent working myocardium. In particular, these studies allowed us to establish the role of T-box transcription factors, as Tbx2, Tbx3, Tbx5, and Tbx18 [16,17,18,19,20,21,22,23,24,25,26], homeobox genes, as Isl-1, Shox2, Nkx2-5, and Pitx2 [27,28,29,30,31,32,33,34], and connexins, as Cx40, Cx43, and Cx45 [35,36,37,38,39,40,41], in ruling gene regulatory networks of the conduction system and/or controlling the interactions of its cells with surrounding tissues. Moreover, fate-mapping analyses based on these transgenic mice and chimeric avian embryos of different species provided insights on the involvement of neural crest stem cells, epicardial stem cells, and second heart field progenitors in the generation of the CCS [29,42,43,44].

Finally, extensive linkage and genome-wide associated studies [18,45,46,47,48,49,50,51,52], post-transcriptional analyses, and next-generation sequencing [49,53,54,55,56,57,58,59,60,61,62] on animal and human tissues and cells, included pluripotent stem cells [63,64], shed light on the ontology of the genes regulated during CCS development and physiology, as well as on their fine-tune silencing provided by specific microRNAs (e.g., miR-1 and miR-128a). Each component of the CCS is, hence, the developmental embryology result of unique hierarchical transcriptional networks, with a prominent orchestrating or ruling role by T-box transcription factors, as previously summarized by Munshi in 2012 [65].

Animal models and ex vivo human autoptic specimens served also to identify and characterize the different anatomical components of the CCS [66,67,68,69,70]. More recently, high-resolution, nondestructive imaging techniques based on phase-contrast computed tomography were applied successfully to the detailed three-dimensional analysis of the human CCS on cadaveric hearts. This allows for us to (i) implement the mathematic modeling to assess its localization and inclination with respect to the whole organ, (ii) estimate the orientation of single cardiomyocytes, and (iii) better predict the propagation pattern of the cardiac depolarization wave [71,72,73,74].

The human SAN is localized in the right atrium at the junction with the superior vena cava and develops caudally towards the inferior vena cava through multiple projections into surrounding tissues. Its size, nearly 15 mm of length and 4 mm of width, and relative content of pacemaker cells (about 10,000) are surprisingly low considering its ability to initiate and propagate the electrical impulse responsible for the contraction of the billions of cells of the whole heart [72,73]. The second CCS station, the AVN, is part of the atrioventricular conduction axis, also comprising the penetrating and branching bundles and the bundle branches of the right and left ventricles. In humans, it is located at the Koch’s triangle apex in the junction between the atrium and the ventricle at the right side of the heart. It connects to the SAN through two pathways at different courses (interatrial septum and crista terminalis) and conduction speeds (fast and slow). The first compact region of the AVN becomes more and more thin reaching the transitional one towards the bundles and branches. Differently from the compact node, these AVN parts show a great interindividual variability [68,69,70,73,75]. Further regions of the CCS are the His bundle and the right and left bundle branches, which extend from the AVN periphery into the membranous septum and then the midseptal area, moderator band, papillary muscles, and, finally, ventricular subendocardium, by following different and specific courses [75]. The electrical path of the CCS terminates in the subendocardium of both ventricles with the large network of Purkinje fibers [76].

All the components of the CCS are strictly coordinated in the cardiac pacemaker function, for which one electrical impulse implies one muscle contraction. SAN innate automaticity generated by pacemaker cells derives from their electrogenic protein profile. HCN1/4 channels, codified by hyperpolarization-activated cyclic nucleotide-gated genes HCN1/4, are responsible for the development of the funny current (I_f_), which is characterized by an inward flux of Na^+^ and K^+^ ions and generated by membrane hyperpolarization. In response to voltage, ERG channels, encoded by the human ether-à-go-go-related gene hERG, develop an outward, delayed rectifier K^+^ current, known as I_k_. In addition, three Ca^2+^ currents are activated: (i) I_Ca, L_, due to voltage-dependent, L-type Cav1.2/1.3 channels; (ii) I_Ca, T_, generated by T-type Cav3.1/3.2 channels; and (iii) I_NaCa_, due to NCX1 channels, whose channels are respectively codified by calcium voltage-gated channel subunit alpha1 C and D genes CACNA1C/D, calcium voltage-gated channel subunit alpha1 G and H genes CACNA1G/H, and sodium carrier family 8 member A1 gene SLC8A1. I_f_ and I_NaCa_ are described as clocks of the pacemaker cells; the first being considered a membrane voltage one, while the second is defined as an intracellular calcium one. The roles and interplay of each clock in the total pacemaking performance of the nodal cell are still strongly debated [77,78].

The cardiac action potential in pacemaker cells starts with the cellular entry of Ca^2+^ through the L-type Cav1.2/1.3 channels. In response to this event, the sarcoplasmic reticulum releases Ca^2+^ ions by the intracellular storage through the ryanodine 2 receptor (RyR2), provoking a change of its intracellular concentration. This induces an upstroke in membrane potential (from −40 to −10 mV) (phase 0), which is not so rapid as when depolarization is mediated by a fast Na^+^ current, and is essential to trigger contraction by myosin-actin interaction. While the calcium channels close, the ERG channels open generating K^+^ influx contributing to membrane potential repolarization to −60 mV, the so-called maximum diastolic potential (phase 3; phases 1 and 2 are negligible in pacemaker cells). The I_f_ channels are, hence, opened in response to this low membrane potential and induce a spontaneous diastolic depolarization. In this phase, T-type Cav3.1/3.2 channels, which are activated at negative membrane potential, play an important role, too [79,80]. The dissociation of Ca^2+^ ions from cardiac troponin in the actin-myosin contractile complex prompts their reuptake by the sarcoplasmic reticulum through SERCA ATPase, as well as their exchange through NCX channels to re-establish ionic homeostasis. I_f_ current alone is responsible to vary action potential from −60 to −40 mV (phase 4), until the threshold is reached to activate I_Ca, L_ channels (−55 mV) and generate a new action potential [81,82]. Electrical impulse is propagated from SAN cells to the AVN ones through atrial connections. A prominent role in the transmission is played by the gap junctions, which are heterogeneously composed by Connexins (Cx) 40, 43, and/or 45, and differently distributed in the CCS depending on age, as well as on the conductance level required in each region [83,84].

Albeit possessing nodal cells, the AVN might serve as the principal pacemaker only during SAN dysfunction. Moreover, its delayed conduction, the so-called AVN delay, protects the ventricles from the uncontrolled spreading of possible atrial hyperexcitability (e.g., fibrillation or tachyarrhythmias). The numerous cellular types composing the AVN are very heterogenous with only the midnodal cells similar to SAN pacemaker ones in terms of HCN4, L-type Cav1.2/1.3, and T-type Cav3.1/3.2 channels. In the peripheral regions, cells organize in multiple, separated bundles with few gap junction connections (specific regional repertoire of Cxs) and voltage-gated Nav1.5 channels (responsible for I_Na_ current), characteristics that likely contribute to the AVN delay onset and maintenance [69,80,85,86,87,88,89].

Purkinje cells are endowed with a clear cytoplasm and a higher expression of connexin 40 when compared to His cells, but do not show the typical electrical isolation observed for the other components of the CCS [76,90,91]. Although this lack of electric insulation, Purkinje cells establish selective communications with each other and with the adjacent ventricular cardiomyocytes through Cx43-mediated junctions [92,93]. Among others, two peculiar currents significantly characterize the electrical activity of Purkinje cells, i.e., I_k1_ and I_Na_ [94], but also calcium handling is distinct in comparison with ventricular cardiac myocytes [95]. Moreover, their sarcoplasmic reticulum lays beneath the cell membrane and surrounds highly packed myofibrils, while T-tubuli are absent [76,96]. These electrical properties are reflected in the action potential, which possesses a long plateau phase but fast depolarization due to the higher excitability of these cells, resulting in the rapid propagation typical of Purkinje fibers [95].

When compared to the pacemaker cells of the SAN, the others found in the rest of the CCS, namely in the AVN and Purkinje fibers, display action potential dissimilarities, particularly in phase 4 of depolarization. This phase that is directly responsible for the heart rate is characterized by decreasing depolarization rates when moving from the SAN cells, depolarizing between 60 and 100 times per minute, to the AVN (40 beats/min) and the Purkinje fibers, whose pacemaker cells can show a 20/min rate. The site-specific characteristics of the CCS eventually lead to different conduction velocities: 0.05 m/s for SAN and AVN, 1 m/s for SAN/AVN connections, 1–3 m/s for His bundle, and 4 m/s for Purkinje fibers. Although governed by SAN in physiological conditions, heart automaticity can be modulated by parasympathetic and sympathetic neurostimulation through acetylcholine and catecholamines, essential to modify cardiac rhythm and pumping activity in response to the changes required during stress and/or body motion [81,82].

All minute electrical modifications and the summation of their effects on cardiac physiology but also pathophysiology were rendered appreciable clinically without any invasive procedure by Wilhelm Einthoven through the string galvanometer. By application of surface electrodes in specific body regions, the extracellular trace of the cardiac electrical activity of each subject, i.e., the electrocardiogram (ECG), can be obtained. In particular, Einthoven recommended the use of three leads connecting (a) the hands, (b) the right hand and the left foot, and, (c) the left hand with the left foot. Each lead allowed to register a peculiar ECG that could provide precise information on the effects of the electrical activity of region-specific cardiomyocytes but also on structural heart changes, as hypertrophy [97]. Nowadays, a standard ECG requires 12 leads, each able to register the extracellular voltage fluctuations, or waves, measured between two electrodes, one positive and the other negative. Current ECGs are developed from the application of six limb leads (I, II, III, aVR, aVL, and aVF) and six precordial leads, for which the positive electrode is positioned on different locations of the chest wall and the negative electrode in the heart center (V1–V6) (Figure 1). Each wave is defined by a peculiar phase of the cardiac cycle in the myocytes of the heart, namely the depolarization of working myocardium in the atria (P wave) and ventricles (QRS complex), the repolarization of the ventricles (T wave), and the papillary muscles (U wave), but it does not directly describe the voltage fluctuations typifying each component of the CCS. Since any wave may vary in its direction and magnitude in the so-called circle of axis in which the heart is central, it is clinical practice to describe it with vector imaging. This latter is an objective modality fundamental in evidencing the multiple clinical signs of rhythm disturbances and advancing in their diagnostic process.

## 3. Heart Rhythm Diseases: Clinical Signs, Underlying Causes, and Modeling Approaches

Any variation of this finely controlled process generating heart rhythm in the CCS might be a cause of profound pathophysiological impairment in the electromechanical activity of working cardiomyocytes and, hence, in cardiac global function, often with life-threatening consequences. Not all voltage fluctuations are pathologic since the heart rhythm does dynamically adjust as a response to physiological needs, as the shortening of action potential and, thus, QT segment, along with the increase of automaticity.

Pathognomonic signs of heart rhythm dysfunction are numerous and can be categorized into mainly automaticity or conduction abnormalities. A disturb in automaticity might occur when CCS pacemaker cells display or acquire alterations in their deputed function or when ectopic foci generate from ventricular cardiomyocytes. Early or delayed afterdepolarizations might induce triggered activity through different mechanisms. Early afterdepolarizations occur in non-pacemaker cells due to alteration in calcium handling in ventricular myocytes, almost resembling diastolic depolarizations; as a result, they may originate one or more extrasystoles, leading to possibly life-threatening forms of ventricular tachycardia. A particular type of the latter is the hazardous torsades de pointes, characterized by a polymorphic QRS complex. Delayed afterdepolarization may be a consequence of calcium overload, which induces the generation of a transient inward current potentially originating spontaneous action potentials [98].

Indeed, the most copious number of rhythm abnormalities are associated with conduction defects. Mechanical stress and myocardial ischemia can create tissue injury and depolarizations causing electrical ionic current instability, as ST-segment elevation or deep Q waves. More frequently, conduction blocks are observed, as (i) first-or second-degree AV block interesting AVN with respectively slow conduction or incomplete atrio-ventricular coupling, (ii) bundle branch blocks with inefficient impulse transmission between His bundle and Purkinje fibers, and (iii) complete conduction block (or third-degree AV block) representing a total electrical dissociation between atria and ventricles. Among the blocks, a further type is the unidirectional one, which is described as the loss of the CCS ability to transmit impulse bidirectionally following anatomopathological modifications. Such a condition is necessary to generate re-entry together with a slower conduction of the action potential and a close conduction loop. The insistence of these conditions is clinically relevant since the concerned area is submitted to a continuous depolarization state giving rise to tachyarrhythmias due to a superimposed rhythm over SAN. Multiple regions of re-entrant activity can co-exist, thus originating fibrillation. The most frequent of this latter in the elders is the atrial form, characterized by a critical outpacing of the SAN and a sustained stimulation of the AVN, which, although exerting its delayed transmission activity, is not able to completely filter some impulses (irregular RR intervals with no detectable P waves at the ECG examination). When the fibrillation regards the ventricles, the condition is more clinically critical due to the loss of coordinated depolarization resulting in blood pump inability. Electrical abnormalities can be generated in the atria also in the case of accessory conduction pathways, as for Wolff–Parkinson–White syndrome, being increased in the epidemiological statistics. Cardiomyocytes—and not Purkinje fibers- compose the fast accessory pathway: as a consequence, the septum depolarizes before the usual physiological time required by the AVN, a sort of pre-excitation that is recognized at the ECG as a delta wave before the Q one. Recurrently, Wolff–Parkinson–White syndrome is associated with re-entry and, consequently, to supraventricular tachycardia, as atrial fibrillation, but also to paroxysmal forms [99,100,101].

Heart rhythm disturbances account for almost 20% of cardiac deaths occurring suddenly. They are very heterogeneous concerning their etiopathogenesis with causes that can be either inherited or acquired. They show variable manifestations, which render often unpredictable the prognosis definition and the anticipation of adverse events. Congenital diseases may eventually manifest during adulthood in response to specific stresses, remaining undiscovered until a dramatic or fatal episode occurs, even in agonistic athletes routinely submitted to ECG to practice [102,103,104,105,106,107,108,109,110,111]. Acquired rhythm pathologies might be induced by several causes, among which drug cardiotoxicity, cardiomyopathies, myocarditis or systemic infections, valve diseases, and interventional/surgical maldeployment of cardiac valve replacements [112,113,114,115,116,117,118,119,120,121,122,123,124,125,126,127,128].

Great progress has been achieved in both clinical diagnosis and management of rhythm diseases [129,130,131,132,133], thanks also to the fundamental knowledge generated through the preclinical modeling of this assorted class of disturbs. Disease modeling has extremely helped to gain more understanding of the causes and the pathomechanistic insights underlying cardiovascular diseases. By utilizing in vitro, in vivo and in silico assays, the modeling goal is to recapitulate typical manifestations observed in humans. In the settings of rhythm disease, in vitro research exploits the heterologous system and cellular lines from humans and other mammals to evaluate at the single-cell level the effects of the dysfunction of cardiac ion channels and other proteins involved in the electrophysiological function. In vivo modeling is based on animals, either naturally bearing a rhythm disturbance or treated to induce it, as through gene engineering (transgenic, humanized, etc.), administration of pro-arrhythmic drugs and chemical compounds, or surgical alteration of the CCS anatomy. In silico models adopt bioinformatic tools to reconstruct and hypothesize the electrophysiological interactions and networks among proteins, cells, and tissues involved in rhythm generation and/or dysfunction, also implementing the evidence gained by in vitro and in vivo investigation and requiring the latter for final validation [134]. As such, at least one of these modeling approaches has been used to mimic diseases on a hereditary or acquired basis, as catecholaminergic polymorphic ventricular tachycardia (CPVT), long and short QT syndromes, Brugada syndrome, arrhythmogenic cardiomyopathy (AC), sick sinus syndrome, atrial fibrillation, hypertrophic, dilated, and left ventricular compaction cardiomyopathies and so on.

In the multiplicity of rhythm disturbances, this review will describe in detail the modeling trials so far realized for some emblematic conditions affecting CCS, as sick sinus syndrome, Brugada syndrome, and atrial fibrillation.

## 4. Sick Sinus Syndrome (SSS): The Complex Modeling of a Multifactorial Disease

Symptomatic SSS is the first cardiac arrhythmia to have an indication for pacemaker implantation due to a series of involved disorders, such as non-physiologic SAN bradycardia, pause, arrest, and exit block, associated with atrial tachy- or bradyarrhythmia, or their alternans (tachy-brady syndrome). This dysfunction of the SAN is more common in the over 65 population, affecting 1 individual in 600 cardiopathic patients with a prevalence in males [135,136,137,138].

Although aging is the primary basis for SSS onset, other intrinsic and extrinsic causes have been identified, as well as some conditions remain idiopathic. Congenital and acquired pathologies altering tissue architecture (coronary artery disease, extracellular matrix disarray, etc.), infective diseases, surgical injury, stress, and, not lastly, familial forms, including muscular dystrophies, channelopathies, and SAN disorders, account for intrinsic causes, while extrinsic causes are more related to drug cardiotoxicity [136,139,140,141,142,143]. The molecular pathogenesis of the SSS is still uncertain, and multifactorial players likely interact to the final pathological phenotype. SSS modeling has proceeded by considering single causes to possibly gain more hints on altered signaling and specific signs, in addition to the electrocardiographic patterns that still remain the unique diagnostic gold standard.

-SSS due to genetic mutations

In families with affected subjects, the main channels to be mutated in SSS are Nav1.5 and HCN, whose genes SCN5A and HCN are responsible for the generation of I_Na_ and I_f_ currents, as observed by genetic screening [144,145,146,147,148,149,150]. It is noteworthy to mention that I_Na_ alterations are not only manifest in SSS, but also in other cardiac arrhythmias and might often generate a mixed clinical picture (as an example, [151,152]). This could be explained by the relatively complex architecture of the channels encoded by SCN5A genes, i.e., Nav1.5, (four homodomains, characterized each by six transmembrane regions with voltage sensitivity (S1–S4) and pore modulation (S5 and S6) [153]), thus mutations occurring in several genetic points could induce protein loss-of-function. Apart for these proteins, other related to the cell electrical activity (e.g., KCNE2, ankyrin-B adapter, PITX2, titin, calcium voltage-gated channels) and nuclear envelope (lamin A/C) were found associated with SSS when in genetic variance or polymorphism [154,155,156,157,158].

In vivo models intended mainly to recapitulate congenital channelopathies. Although interspecies differences exist with rodents, transgenic mice carrying the same mutations in the genes codifying for aforementioned channels display a similar phenotype with humans, also correlating with sex and aging. By the targeted disruption of SCN5A gene in mouse, Lei et al. showed in 2005 that SCN5A heterozygosis (Scn5a^+/−^) generated mice displaying typical SSS signs, as SAN bradycardia, conduction, and exit block. Moreover, they shed light on the importance of I_Na_ current—generated in the adult life from peripheral SAN cells [159,160] for atrial activation [161]. Further research by Jeenavaratnam et al. in 2010 also evidenced the sex-dependent effects of SCN5A heterozygosis on sinoatrial function in aging, male mice, which displayed longer RR intervals, especially with prolongation of ventricular repolarization [162]. Through Scn5a^+/−^ mice, Hao et al. [163] demonstrated that electrophysiological alterations were accompanied and progressively aggravated by structural fibrotic modifications, due to upregulation of TGF-beta1 and consequent increase in collagen synthesis.

The electrophysiological observations in these mice and/or their excided hearts were also investigated in cell lines defective for SCN5A gene, specifically in the rabbit. After the early analysis of I_Na_ current generation in post-natal rabbit SAN [159], Butters et al. demonstrated in vitro by two-dimension (2D) cell-based voltage-clamp assays that the isolated cells from Scn5a^+/−^ animals displayed different pacing abilities when moving from the central node towards the periphery, with the latter characterized by a dramatically reduced activity [164]. These studies also included in silico reconstructions with simulation of I_Na_ current modifications during aging and disease starting from experimental data.

It must be not surprising that HCN genes are found mutated in SSS due to their relevance in I_f_ generation in SAN [78,165]. Fenske et al. evaluated the effects of HCN1 gene deficiency in knockout mice. Microarray and immunohistochemical analyses on isolated HCN1^−/−^ SANs revealed that apart from the null HCN1, ion channels and proteins were unvaried in their molecular and protein expression. However, SAN cells showed decreased beating rate, alterations in the beat-to-beat interval, and delayed impulse formation during in vitro microelectrode experiments. These observations related to in vivo outcomes collected from ECG and intracardial electrophysiology: prolonged sinus node recovery time, RR interval, high beat-to-beat dispersion, and sinus pauses were documented in HCN1^−/−^ mice, as indicative signs of SSS. By studying spontaneous sinus cycle lengths, they reported that the range between the shortest and longest values did not change in these mice with respect to wild-type animals, although the average values were approximately duplicated *ab initio*. No compensatory mechanisms were instigated and bradycardia was maintained, thus revealing the relevant role of HCN1 in the generation of a basal depolarizing current [166]. Although HCN1 is pivotal for I_f_ functionality, its gene is generally not found mutated in humans, where the predominant mutations regard HCN4. Heterologous expression systems have been widely adopted to analyze the behavior of HCN4 mutated proteins. Cell lines of COS-7 and HEK-293 were transfected with several HCN4 mutated genes detected in patients. In 2004, Ueda et al. expressed in COS-7 cells a rabbit HCN4 mutated gene, similar to a human one reported in an SSS-affected patient (sinus bradycardia, cardiac arrest, polymorphic ventricular tachycardia, and *torsade de pointes*). This mutation was located in an HCN4 region connecting two domains, one the core transmembrane domain and the other a highly conserved one for all HCN protein family. Patch-clamp analysis on transfected COS-7 cells evidenced that the mutation did not affect channel voltage-dependence. Indeed, activation occurred faster, while deactivation was slower when compared to wild-type cells [167]. As pointed out by Verkerk and Wilders [168], nor in vitro 2D assays neither in silico simulations generated might be fully able to reproduce the diseased phenotype observed in humans, particularly for some mutations. In 2018, Moeller et al. expressed two newly discovered missense HCN4 mutations of two young SSS patients, as well as one already known, in *Xenopus laevis* Stage V oocytes and COS-7. In homo- or heteromeric combinations of derived proteins, the mutations were revealed to induce complete loss-of-function following a substantial decrease in HCN4 membrane expression. A computational model was generated by gathering together the data collected by the study of activation and deactivation kinetics of mutants and wild-type channels. The derived molecular dynamics allowed us to hypothesize a possible structural channel alteration induced by new mutations that might have interfered with proper trafficking and voltage sensor functions [169].

From preclinical studies, it emerged that SSS pathomechanism(s) might be more complex than a deficit of a single channel or protein. The feedback between Ca^2+^ handling and membrane potential clock might result also as altered, as shown in vivo by Torrente et al. in an NCX^−/−^ mouse model and confirmed in silico by Morotti et al. simulating pathological modifications of intracellular Na^+^ [170,171].

With the consistent evidence of the importance of desmin and desmosomal proteins for SAN functionality [172,173], it is also emerging a possible causative role of desmoplakin mutations not only in the etiogenesis of AC but also of SSS. Battipaglia et al. observed an association between a reduced heart rate and AC [174]. Recently, lines of induced pluripotent stem cells lines (iPS) from a patient carrying a desmoplakin mutation and diagnosed for SSS and progressive conduction disorders were generated [175]. Patch-clamp analysis on iPS-differentiated cardiomyocytes revealed impaired Nav1.5 and L-type Ca^2+^ channel dysfunction [176]. Similar studies on iPS-differentiated pacemaker cells could allow us to gain more information on the altered electrical phenotype induced by desmoplakin mutations.

Although not observed in human patients, other mutations were found associated with SSS in animal models: transient Notch activation induced typical pathological signs in transgenic mice. As shown by Qiao et al. [177], a temporary Notch activation may induce long-term molecular alterations in the genes essential for SAN development and electrical activity, as Nkx2.5, Tbx3, Tbx5, but also Scn5a, and Gja5 (codifying for Cx40), by creating an arrhythmogenic substrate.

-SSS due to aging

Apart from genetic mutations, a reduction in the expression of proteins fundamental for the current generation has been observed during aging. It is hypothesized to be correlated to the SSS so frequently diagnosed in the elders. As for other tissue and organ deteriorations occurring with aging, it could be likely due to an epigenetic mechanism. Alghamdi et al. constructed an action potential mathematical model of the SAN cell of an aging rat by gathering together outcomes from different studies in the literature. They demonstrated that SAN dysfunction could arise from concomitant, age-dependent alterations in several currents and electrical functions, including Ca^2+^ handling [178].

-SSS due to inflammatory conditions and structural alterations

Inflammation and structural modifications, both congenital and acquired, might destabilize SAN physiological behavior and lead to an irreversible shift towards electrical dysfunction.

At the cardiac level, inflammation might establish through several pathological disorders. Hypoxia and oxidative stress occurring in cardiovascular diseases, as cardiac ischemia/infarction, coronary artery disease, atherosclerosis, and eventually heart failure, generate alterations in numerous molecular pathways, with critical functional and structural sequelae [179,180,181,182,183,184,185,186,187,188]. Induced cardiac injury is particularly associated with higher levels of catecholamines, G protein-coupled receptors, and angiotensin II. Both catecholamines and angiotensin II are well known to specifically modulate SAN chronotropy or heart rate in a species-specific manner through G protein-coupled receptors, as ß-adrenergic, muscarinic, and AT_1_/AT_2_ receptors; moreover, they interact with each other with indirect effects on the SAN (a recent overview of the SAN neurohumoral control has been proposed by MacDonald et al. [189]). Hence, an overload in stimulation during stress conditions has serious consequences on SAN and global heart electrical activity [190], as in the case mediated by G-protein coupled receptors. The signaling of these proteins is negatively modulated by regulator of G protein signaling (RGS), a class of GTPase-accelerating proteins (GAPs). Through seminal work adopting different technologies (cardiomyocytes differentiated from embryonic stem cells (ES) and knock-in mice), Fu et al. investigated the RGS role in heart automaticity after suppressing the binding and inhibition of these GAPs on several G_alpha_ proteins. RGS-insensitive mutant ES-differentiated cardiomyocytes did not show any variation in heart beating by isoproterenol stimulation, although an important bradycardia followed adenosine A1 and/or muscarinic M2 receptor agonist administration, depending on the introduced mutation [191,192]. In addition, isolated, perfused heart from knock-in mice in homo- or heterozygosis (respectively, GS/GS and +/GS) demonstrated increased sensitivity to carbachol, with signs of sinus bradycardia, as well as third-degree AV block [193]. Taken together, these results reveal the powerful reduction effect of endogenous RGS on G_alpha_ proteins and, hence, the ability of RGS to control SAN automaticity independently from the neurohormonal stimulation by the nervous system.

Mitochondrial oxidative stress has been demonstrated to induce alterations in the expression of HCN4. Indeed, the deletion of thioredoxin 2 in the whole mouse heart generated not only dilated cardiomyopathy and AV block but also sinus bradycardia, following HCN4 downregulation through HDAC4-MEF2C signaling pathway modifications. These mice displayed, in fact, a sensibly reduced expression of HCN4 in SAN cells and typical electrophysiological signs of SSS [194].

Myocardial ischemia/reperfusion injury after acute myocardial infarction encompasses oxidative stress, Ca^2+^ overload, and inflammation and is associated with SSS. Animal models of SSS can effectively be generated by clamping the SAN region with hemostatic forceps, as shown by Zhang et al. in rats [195]. The surgical procedure impacted SAN anatomical cell distribution and significantly reduced HCN4 and SCN5A molecular expression, as well as Ca^2+^ handling, leading to decreased heart rate and long RR intervals. This model was used by the same group to evaluate the effects of a herbal pharmacological compound, Zenglv Fumai granule, in rescuing SSS phenotype, showing the efficacy on pathways related to myocardial ischemia/reperfusion injury, as TRIM genes [196]. Analogously, they confirmed these outcomes in an in vitro 2D model based on human cardiomyocytes AC16 submitted to hypoxia and reoxygenation [197]. Liu et al. analyzed the effects of ischemia and reperfusion injury on rabbit SAN cells and observed a remarkable reduction in the I_f_ current density. They also evaluated the consequences of the treatment with the herbal medicine astragaloside, which upregulated HCN4 molecular expression and protected cells from the onset of stress responses, as increased production of stress fibers [198]. A multiscale, 2D simulation of ischemia-mediated SAN-atrium dysfunction was created by Bai et al. [199] by loading experimental results generated on the ionic currents of rabbit cells. I_NaCa_ and I_K_ resulted to be impaired in this model in ischemic settings with a reduction in SAN heart rate and atrial conduction velocity. Interestingly, simulated acetylcholine vagal stimulation worsened both ischemic consequences and SAN dysfunction, leading to the arrest and exit block of the latter.

-SSS due to drug or chemical cardiotoxicity

Several pharmacological compounds, included antiarrhythmic drugs, have demonstrated cardiac toxicity targeted on the SAN and, thus, show the ability to induce or worsen SSS. Just as a few emblematic examples, the calcium antagonists verapamil and diltiazem exert inotropic effects and might prompt other conduction disorders, as sinus rhythm reversal [200], or the sodium channel blocker propafenone is an effective antiarrhythmic that, however, might provoke SAN block in patients affected by SSS [201].

The anticancer/antibiotic agent streptozotocin is largely used in experimental diabetology for its ability to provoke selective cytotoxicity in pancreatic islet ß-cells. Streptozotocin has also been demonstrated to induce SSS in rodents following hyperglycemia. Diabetes mellitus has also been reported as a possible risk factor for SSS in humans, most likely due to downregulated electrical signaling, oxidative stress, inflammation, atrial fibrosis, and other structural alterations induced by this metabolic disease [202]. As shown by Kondo et al. [203], streptozotocin-induced diabetes provokes systemic inflammation, high levels of ROS, and apoptosis in cardiac fibroblasts, as well as specific signs of SAN dysfunction, as decreased expression of HCN4, increased expression of TGF ß-1, fibrosis, macrophage infiltration, and reduced baroflex sensitivity. Intriguingly, this complex phenotype could be reversed by administering the anti-inflammatory molecule IL-10.

Angiotensin II might also be used to mediate SSS in animals, thus simulating the possible overload observed in patients. Through this modeling, Zhong et al. showed in 2018 that the transient receptor potential subfamily M member 7 (TRPM7), implicated in cardiac fibrosis, has a role in SAN fibrotic lesions via the activation of the angiotensin II/Smad pathway. They suggested, moreover, that TRPM7 could be further investigated as a pharmacological target to prevent the progression of SAN fibrosis [204]. Just recently, Zhang and colleagues investigated the effects of an herbal medicine Shenxian-Shengmai oral liquid in a SSS rat model, induced by the infusion of angiotensin II through an osmotic pump. They also developed an in vitro 2D model using HL-1 atrial cardiomyocytes, which they considered as surrogates of SAN cells. They found that angiotensin II mediated in cells the upregulation of PKC/NOX-2 signaling and provoked the characteristic molecular changes also in vivo: among others, reduced levels of HCN4 and HDAC4 were observed. The administration of Shenxian-Shengmai generated protection from oxidative stress both in vitro and in vivo, rendering less severe the signs provoked by angiotensin II [205].

Other SSS animal models might be generated administering cytotoxic substances, such as formaldehyde and sodium hydroxide, in the SAN region. This technique, generally referred to as pinpoint press permeation, was applied to study SSS in rabbits and mice and consists in applying a soaked cotton ball on the SAN region until sinus rhythm is progressively lowered. In the comparison with the previously mentioned surgical clamping, Zhang et al. tested the efficacy of 10% sodium hydroxide to induce SSS in rats by using the internal jugular vein as the administration route. The infusion was able to reduce automaticity less importantly than the surgical procedure. However, sinoatrial recovery and conduction times were significantly prolonged in all two groups, thus confirming the suitability of the injection technique in generating SSS signs [195]. As documented by Roh et al. in rabbits [206], the molecular changes associated with the local administration of 20% formaldehyde provoked a severe atrial remodeling, starting from important alterations in the molecular signature of inflammation and extracellular matrix homeostasis and culminating with the onset of fibrosis.

A summary of the SSS modeling strategies and implications for mechanistic insights and clinical translation is provided in Table 1.

## 5. Brugada Syndrome: Modeling a Rhythm Disorder with Still Incompletely Defined Etiology

Brugada syndrome (BrS) is a life-threatening condition characterized by ventricular tachycardia/fibrillation and increased risk of sudden cardiac death. In a structurally normal heart, rhythm dysfunction has been found mainly associated with mutations in the SCN5A gene. As before mentioned, the targeting of this gene is commonly observed in some forms of familiar SSS and LQT, so that affected patients may manifest the signs of both diseases. The inheritance of BrS is, in reality, more complex. Mutated genes are not only SCN5A, but also other codifying for other sodium channels or proteins, although the occurrence of these mutations is described rarely in the clinics. Most mutations distribute as autosomal dominant traits, penetrance might be variable, and the male sex shows increased association. To complicate the diagnosis, cases of positive phenotype might be observed also in family members with negative genetic screening, but carrying single nucleotide polymorphisms [151,207,208,209]. A recent review by Monasky et al. summarized the complex genetic scenario of BrS and questioned the initial definition as Mendelian disease. In fact, it is consistently clear from next-generation sequencing studies that BrS can be classified as genetically inherited specifically for SCN5A mutations [210].

All mutations lead to typical signs at the ECG examination: the heart of BrS-affected patients displays a coved type ST-segment elevation with negative T wave in the right precordial leads and right bundle-branch block. Actually, ventricular fibrillation manifests mainly in the right cardiac side at the level of the outflow tract [211]. Interestingly, some structural variations of this tract seem to predispose to BrS. Increased collagen, fibrosis of the epicardial surface and of the *interstitium*, reduced expression of Cx43, and, hence, decreased number of gap junctions have been found as associated and concurring to create the arrhythmogenic substrate exiting in reduced contractility and BrS [212,213]. However, the molecular pathomechanistic basis leading to BrS phenotype is still incompletely elucidated. The symptomatology may manifest in adulthood variably in age and penetrance and this prevents the acquisition of accurate estimations of prevalence, as well as the realization of large genetic screening to better understand the causal triggers.

The effects of SCN5A mutations have been widely investigated through animal models. To better mimic the clinical picture, concurrent genetic variants or polymorphisms clinically observed in other genes, as MOG1 and TRPM4, were also modeled.

-BrS due to SCN5A haploinsufficiency

Transgenic mice haploinsufficient for Scn5a gene, the homologous for the human SCN5A, were generated by knockout technology for the first time by Papadatos et al. in 2002. In vitro patch-clamp analysis on Scn5a^+/−^ mouse ventricular cardiomyocytes revealed a 50% reduction of I_Na_ current without any change in channel gating properties, while slowed conduction velocity, but normal QT interval was observed in vivo at the ECG when compared to wild-type animals. This considerable decrease in sodium conductance was hypothesized to explain the abnormal electrical activity noted in vitro in isolated ventricular preparations, after that Scn5a^+/−^ mouse hearts were freed from atria and, thus, from their electrophysiological control. Conduction block, re-entrant arrhythmias, and ventricular tachycardia were confirmed in these ventricles, all signs reported in humans, too, and possibly underpinned by reduced tissue excitability. Similar to the clinical observations, wide variability in sign manifestation was detected from all performed characterizations. Intriguingly, double knockout mice for Scn5a could survive during the *in uterus* life no more than E10.5 gestational day and manifested critical structural cardiac alterations that were described as severe ventricular morphogenetic defects [214].

With the same transgenic model, Leoni et al. further investigated the commonalities of Scn5a haploinsufficiency and human BrS phenotypes. The prolongation degree of QRS interval was used to identify two Scn5a^+/−^ animal subgroups. This selection criterion was revealed to be successful to distinguish the two subgroups based on disease severity since aging was not changing the QRS duration over time, although it was found to aggravate the conditions in animals manifesting particularly prolonged QRS at the baseline. Episodes of ventricular tachycardia were observed only in old haploinsufficient mice showing importantly increased QRS interval. Sign exacerbation was also due to an important fibrotic state occurring in the heart of this animal group. All these findings are also observable in humans. Molecular analyses on the hearts of the two animal groups evidenced a correlation between severity and I_Na_ current reduction: similar levels of Nav1.5 mRNA were quantified in all Scn5a^+/−^ mice but protein expression was significantly lower in animals with more prolonged QRS [215].

Matthews et al. proved that the wavelength restitution could be used as a prognostic marker of action potential duration. In a Langendorff ex vivo perfusion system, they recorded action potential variations in right and left ventricles, as well as epicardial and endocardial surfaces in response to a dynamical increment of pacing. These analyses revealed that maximum conduction velocity was higher in the endocardial surface than the epicardial one of haploinsufficient right ventricles. This conduction block could be worsened by pharmacological stimulation with fleicanamide and quinidine. By using recovery wavelengths, a parameter obtained by the multiplication of conduction velocity and diastolic interval, they showed in addition that more effective prediction could be acquired on alternans magnitude and, therefore, on arrhythmia severity, instead of considering the sole diastolic interval [216].

Kelly et al. deeply investigated the possible structural variations occurring in the right ventricle of Scn5a^+/−^ mice. Optical recording of action potentials by two-photon microscopy during 7 Hz pacing of isolated hearts perfused ex vivo exposed intolerance by transgenic mice. Preclinical signs observed in previous studies were re-confirmed. Remarkably, the transmural gradient in action potential duration, evident between the right and left ventricles in wild-type hearts, completely cleared in Scn5a haploinsufficiency. Additionally, transmural conduction velocity was lower in both ventricles of transgenic hearts, with a significant decrease on the right side, accompanied by increased dispersion. By augmenting the frequency of pacing, delays in activation were only observed in the right ventricles of Scn5a^+/−^ hearts. From an anatomical point of view, right ventricles showed a higher number of non-vascular regions than the left ventricles, and this difference was apparent independently of the mutation. The abnormal electrical behavior observed for Scn5a^+/−^ hearts could be reproduced also in wild-type organs by administering tetrodotoxin, a highly specific voltage-gating sodium channel blocker [217]. As such, these physiological anatomical dissimilarities between right and left ventricles contribute de facto to create a structural substrate for BrS manifestation when Nav1.5 channels are mutant.

Taking into consideration that ventricular fibrillation episodes are recurrent at night during sleep when vagal stimulation is reduced, Finlay et al. explored the effect of neurohormonal stimulation on Scn5a^+/−^ mice. Electrode array assays were performed with electrical stimulation at 5 Hz on hearts isolated from old mice and perfused in a Langendorff system, and cardiac tissue preparations from excided organs from young animals. Reduced conduction velocity was observed in these independent experiments. In both settings, carbachol (10 µM) and isoprenaline (100 nM) were administered to simulate vagal and sympathetic stimulations, respectively. Longer effective refractory periods were documented for Scn5a^+/−^ cardiac organs and tissues. The latter did not respond to isoprenaline and/or carbachol, differently from wild-type counterparts that augmented or decreased conduction velocity, in a more consistent manner for cardiac slices. Premature extrastimuli doubled the conduction delay in Scn5^+/−^ hearts in comparison with wild-type conditions [218]. These results are suggestive of a loss of the ability to reply to autonomic stimulation in the case of Scn5a haploinsufficiency. The causes underneath this epiphenomenon were not investigated in this study. Indeed, the clinical challenge of BrS patients with a radioactive norepinephrine analog evidenced a reduced uptake, suggestive of a presynaptic impairment of cardiac sympathetic innervation [219].

The potential weakening of the functional liaison of Nav1.5 and Kir2.1-3 channels was investigated by Perez-Hernandez et al. in Scn5^+/−^ mice. By using this model, they demonstrated a reduction of I_Na_ and I_K1_. Not only Nav1.5 mutant proteins were responsible for this concurrent decrease but also a mutual effect could be established by Kir 2.X channels. They investigated the dynamic changes of I_Na_ and I_K1_ currents in mice carrying the knockout of Scn5a and Kcnj2OE genes. The latter codifies for Kir 2.1 channels and when in haploinsufficiency, it leads to a protein overexpression. Kir 2.1 overexpression increases I_K1_ but also I_Na_ in Scn5a^+/−^ mice, with rescuing effects on the mouse BrS phenotype [220].

-BrS due to SCN5A point mutations

Nevertheless, Scn5a haploinsufficiency by targeted gene disruption does not represent a real human scenario. In fact, human Nav1.5 proteins were found mutated due to defective gating or trafficking properties, as well as for haploinsufficiency determined by premature transcriptional stop signals [221]. Therefore, models were also specifically selected to reproduce mutations typically diagnosed in BrS patients.

Remme et al. used Cre-LoxP-mediated gene targeting to deeply investigate in mice the electrophysiological pattern given by the human 1798insD SCN5A mutation found in a large Dutch family as causative for several rhythm disturbances, as SAN dysfunction and BrS. At first, they observed that mice homozygous for the mutation could not survive. Heterozygous animals did not show any difference when compared to wild-type ones apart from significantly prolonged PQ, QRS, and QTc intervals. These alterations, especially for PQ, could be exacerbated by flecainide administration, which caused the appearance of sinus bradycardia and/or arrest. After heart stimulation, total activation time was increased in the right ventricles, but not in the left ones. A similar trend was observed for an effective refractory period. Cardiomyocytes derived from mutated animal hearts displayed longer action potential, already when stimulated at 2 Hz, and reduced upstroke velocity at −120 mV [222]. This model was efficiently able to reproduce the predominant RV affection, the prolonged effective refractory period, SAN dysfunctions, and the lack of ventricular arrhythmias displayed by family carriers of 1798insD mutation.

Keller et al. first introduced in 2005 the concept of dominant-negative effect in BrS diagnosis, by identifying a new mutation inducing L325R protein variant. This latter gave rise to a reduction in Nav.1.5 channel number and generated current when overexpressed in a 1:1 ratio with wild-type SCN5A gene in HEK293 cells. They also investigated in vitro the effect of fever occurring in the BrS proband. A computational model was generated that demonstrated premature repolarization and dome loss in the action potential of cardiomyocytes holding a mutation-reduced I_Na_ current and exposed to 40 °C [223].

In 2012, Clatot et al. reported the modeling of the SCN5A mutations originating R104W and R121W mutants of the N-terminal region. These mutations were diagnosed in three distinctive subjects affected by BrS. In particular, the R104W carrier did not present symptoms but the classical ECG signs of ST elevation, inverted T waves, and partial right bundle branch block. The unique R104W expression in the heterologous system HEK293 and in rat neonatal cardiomyocytes led to a complete suppression of I_Na_ current. The mutant could not reach the membrane as in wild-type settings and was found to localize at the perinuclear and intra-cytoplasmic regions, suggestive of trafficking impairment. When the R104W-inducing variant was co-transfected at 1:1 ratio with the wild-type gene in HEK293, the decrease of I_Na_ current was less critical (−80%), revealing a dominant-negative effect of the mutation. Clatot and colleagues demonstrated that this was specifically due to the defective N-terminal after they co-transfected HEK293 cells with the sole N-terminal and its variant ΔNter, where the codifying region was deleted [224]. The same group evaluated in mice the effects of the Nav1.5-R104W variant by adopting viral DNA recombination and trans-splicing for animal engineering. In particular, they systemically injected adeno-associated viral vectors carrying the human mutated SCN5A gene. Overexpression of R104W channels in vivo induced lowered heart rate, dramatic reduction of I_Na_ current, prolonged RR intervals, and longer P-wave duration, indicative again of the dominant-negative effect observed previously in vitro [225].

Interestingly, Wang et al. further explored the work of the group performed in 2005 by hypothesizing a possible binding site for calmodulin in the N-terminal region of Nav1.5 channels. They tested this hypothesis by considering several N-terminal mutants, included R104W, R121W, and a new mutated protein, Y87C, described in a Russian family. Both R104W and Y87C mutations were (re)confirmed as causative of a dominant-negative effect when expressed in the transformed human kidney TsA-201 cell line. Moreover, the derived mutant proteins showed binding to calmodulin, as well as wild-type Nav1.5 channels, as observed in the COS7 cell heterologous expression system. Conversely, the R121W mutation did not produce any dominant-negative effect and coded protein did bind to calmodulin in a weak manner [226], thus evidencing the essential role of the R121 site for calmodulin interaction and the correlation of the latter with dominant-negative effect. Although the study reported the novel insight on the binding of calmodulin to Nav1.5 channels specifically at the N-terminal domain, it remains to be elucidated whether similar findings can be proved in a more physiological model, such as cardiomyocytes derived from primary culture or differentiation of pluripotent stem cells.

As also confirmed by this review, mice have found a large use to model rhythm disturbances. Easy handling due, for example, to the small size and gestation time, but also the belonging to mammals, and more established techniques for their genetic manipulation have contributed to consider the mouse the first animal choice in biomedical research. However, it must be contemplated that cardiac anatomical and (electro)physiological differences between mice and humans are consistent and might prevent the real recapitulation of patients’ diseases. In order to overcome these flaws and generate a mammalian, large-size model with larger similarity with humans, Park et al. genetically humanized a Yucatan minipig to express the E558X-inducing mutation, previously identified in a pediatric BrS patient. The introduced mutation did not result in any structural effect on porcine hearts. Although no events of sudden death occurred during the two years of follow-up, the engineered animals displayed at baseline several electrophysiological alterations, as slow conduction, prolonged P wave, PR, and QRS intervals at the ECG, as well as sustained atrial-His and His-ventricular conduction periods at intracardiac analysis. Moreover, SAN recovery time resulted to be prolonged and hearts displayed decreased velocity in conduction. All these electrophysiological signs were hallmarks of the rhythm disease observed in humans and were confirmed at the protein expression level in Western blotting by a significantly reduced Nav1.5 channel density in atrial myocytes, which corresponded to decreased I_Na_ current. The electrophysiological pattern was aggravated by temperature rise and signs observed in young transgenic pigs worsened with animal aging, completely parallel to the human phenotype. Differently from other BrS mutations, no differences in mutated Nav1.5 expression were noted between the right and left sides of the heart. Indeed, no specific modifications were observed during animal aging at the macroscopic level in terms of myocyte hypertrophy, fibrosis, size, and general function, and at the microscopic level regarding the molecular rearrangement of the proteins interacting with Nav1.5 channels [227].

An even more physiological model to study BrS is represented by the in vitro platforms based on cardiomyocytes differentiated from patient-specific, human iPS. After the advent of human iPS technology in 2007 [228], the earliest pluripotent cell lines carrying BrS-inducing mutations were generated 5 years later by Davis et al., who first recapitulated on a dish the phenotype of the human 1795insD SCN5A mutation. The latter was previously identified and studied in transgenic mice by Remme et al. [222]. The proof-of-concept of the suitability of pluripotent stem cells to model BrS was gained in this novel study. ES, employed to generate the previous mouse model, and iPS, derived from both the same mice and human patients carrying the SCN5A mutation, were efficiently differentiated into cardiomyocytes in vitro by coculturing with END-2 cells and patch-clamp analyses were carried out. Cardiomyocytes derived from the different pluripotent stem cells carrying the SCN5A mutation display reduced I_Na_ current, decreased upstroke velocity, and prolonged action potentials, which deteriorated with increasing pacing [229]. More recently, a rescuing approach of the diseased phenotype was successfully realized in these mouse and human iPS models by Portero et al. with the administration of the late sodium current inhibitor GS-458967 [230].

In 2016, Liang et al. generated iPS lines from 2 patients with a BrS diagnosis, both with a familiar history of sudden death. Different mutations were identified in these subjects: for one, a double missense mutation leading to R620H and R811H, the latter known to be located in a site fundamental for gating properties, while for the other, a base pair deletion, resulting in a premature channel truncation. The action potential of cardiomyocytes, differentiated from these two patient-specific iPS cells, was abnormally prolonged, with enlarged variability of the peak-to-peak interval, and slower depolarization. In addition, I_Na_ current appeared decreased, consistent with a lower membrane expression of Nav1.5 channels. In iPS-cardiomyocytes from BrS patients, Ca^2+^ handling was altered with reduced transient amplitude and maximal intracellular concentration. The molecular analysis of these cells confirmed the downregulation of Ca^2+^ handling pathways, but also of cardiac hypertrophy and ß-adrenergic stimulation. Intriguingly for translational purposes, targeted genome editing of patient-specific iPS by CRISPR-CAS9 technology corrected the mutation and rescued the BrS phenotype [231].

Selga et al. developed a human iPS cellular model from the dermal fibroblasts of a patient carrying the R367H-inducing SCN5A mutation and compared it with a heterologous expression system in tsA201 cells. No gross changes in morphology were observed for iPS-cardiomyocytes derived from the patient when compared to a control. However, electrophysiological properties resulted to be altered for the first when submitted to the whole-cell current recording after the application of two distinct differentiation techniques, i.e., embryoid bodies and cell monolayer. I_Na_ current was reduced by more than 30% in peak density. In addition, in voltage-dependence conditions, an increase in the activation and a decrease in steady-state deactivation were registered, together with an accelerated channel recovery from inactivation, a proposed mechanism for arrhythmia generation in BrS patients. Parallel studies in tsA201 cells revealed that the mutant protein could reach and was located in the membrane during either heterozygous or homozygous expression, therefore eliminating any hypothesis of dysfunctional trafficking induced by the gene mutation and favoring instead a possible loss-of-function to support I_Na_ current density decrease. The human iPS line presented, however, a higher potential to mimic the disease by harboring the information of all the genetic background of the patient, not only of the mutated gene [232].

Being patient-specific, iPS-cardiomyocytes provide the distinctive genetic background of the subject and thus should be a superior system than other BrS mammalian models, which failed to show the loss-of-dome typically observed in the clinic (as an example [214,222,227]). Unfortunately, these cardiomyocytes are known to be incompletely mature after the iPS differentiation protocols applied so far. The absence or low density of protein and/or specific currents can represent a significant obstacle to effectively mimic the electrophysiological hallmarks of BrS-affected native cardiomyocytes. De facto, the resting membrane potential of iPS-cardiomyocytes is 30 mV higher than their native counterpart, due to an absent I_K1_ current. This, in turn, induces a limited I_Na_ current due to the rapid inactivation of its fast component. At the same time, this depolarized resting membrane potential inactivates I_to_. This is the reason why iPS models of BrS were able to evidence some differences in I_Na_ current when compared to healthy similar systems, but could not characterize phase-1 repolarization, generally absent in their differentiated cardiomyocytes [231]. To overcome this limitation, Ma et al. investigated the effects of a compound SN5A mutation leading to A226V and R1629X mutants in an iPS model generated from a BrS patient with forcibly established I_K1_ current. The heterologous tsA201 cell system was adopted, too. Western Blot analyses of modified tsA201 clarified that trafficking impairment was induced within R1629X mutants and resulted in complete I_Na_ abolishment, as observed by the current recording. On the other hand, A226V mutants displayed a normal Nav1.5 protein level but half I_Na_ current, without any consequences for steady-state inactivation and activation. A 25% I_Na_ reduction was also appreciated in patient-specific iPS-cardiomyocytes, consistent with a decreased transcription of mutated SCN5A. These cells strongly differed in the rate of recovery from inactivation with respect to a control line obtained from a healthy sibling. The missing I_K1_ current was established by inserting a synthetic one in the membrane of iPS-cardiomyocytes using a dynamic patch clamp. A dome-like pattern was visible during the phase-1 repolarization of the action potential in I_K1_-modified control iPS-cardiomyocytes. BrS counterparts displayed a rounded hint, with an important reduction of maximum upstroke velocity and action potential amplitude, coherent with a significant decrease of I_Na_ current. An increased prolongation of the action potential was registered at decreasing pacing, with 0.1 Hz sufficient to push 75% of iPS-cardiomyocytes to this behavior. This pattern was aggravated by temperature rise from 24 to 34 °C. All these observations acquired through this BrS patient-specific iPS system were in favor of the hypothesis of repolarization disorder already suggested by Aiba et al. in their in vivo canine model (ref. [233], which will be reviewed in next subsections) and phenocopied the rhythm alterations detected in BrS patients at rest and/or with fever. Ma et al. demonstrated, indeed, that similar findings were not revealed by means of another iPS line, generated with genome editing to insert a mutation giving rise to a T1620M mutant with a milder effect on I_Na_ current [234].

Intriguingly, CRISPR-CAS9 was applied also to edit the human iPS line from healthy to A735V SCN5A mutant, as proposed by Angsutararux et al. in 2019 [235]. Later, De la Roche et al. compared this iPS model to the heterologous HEK293 cell system. Wnt-modulated differentiation of iPS harboring the mutation allowed to obtain cardiomyocytes that expressed the same density of Nav1.5 channels as the control cells but showed decreased upstroke velocity of the action potential and a positively shifted curve of voltage-dependence of activation. A similar effect on the activation curve was observed in HEK293 cells transfected with mutated SCN5A gene. The heterologous system allowed also to exclude any hypothetical impairment in trafficking and confirmed the alteration of channel properties induced by the mutation [236]. Once again, the adoption of independent modeling approaches demonstrated its validity to provide complementary insights on the studied disease. Further proof is given by the study by Perez-Hernandez et al. previously mentioned regarding Scn5^+/−^ mice. In fact, they used iPS and rat cardiomyocytes besides the transgenic mice in order to analyze the expression of Nav1.5 mutants due to R878C-inducing mutation and proved a gating defect by excluding improper trafficking [220].

-BrS due to mutations in other genes

Other members of the SCN gene family might undergo mutations clinically described as associated with BrS. Two variants of the SCN10A c.3803G>A and c.3749G>A were identified by El-Battrawi et al. in a subject diagnosed with BrS. Human iPS were generated from dermal fibroblasts and differentiated into cardiomyocytes. The latter showed an importantly increased expression of SCN5A (10-fold) and SCN10A (2-fold). This concomitant increase of both SCN transcripts can be explained by the role of SCN10-encoded protein, which serves to modulate Nav1.5-related I_Na_ current in its late phase and heart conduction. By means of patch clamp analysis, peak I_Na_ current resulted to be reduced in patient-specific iPS-cardiomyocytes. Analogously, peak I_Ca,L_ and I_NaCa_ were negatively impaired by SCN10A mutation. Additionally, the amplitude of the action potential and the maximum depolarization velocity were reduced. An increase in the number of early and delayed afterdepolarizations was documented, too [237].

Other genes, that were found varied or mutated in BrS patients, generally play a role in the function of Nav1.5 channels. Ozhathil et al. focused on the modeling of the mutations in the TRPM4 gene. TRPM4 codifies for a Ca^2+^-activated, non-selective cation channel expressed in several tissues. Its deletion in mice led to intraventricular conduction defects observed at the ECG and reduced I_Na_ current verified at perforated patch-clamp [238].

MOG1 protein is fundamental for the correct distribution of Nav1.5 channels at the plasmatic membrane and the mutation leading to E83D protein variant was reported in a BrS patient. Moreover, MOG1 overexpression could have a central translational impact for the treatment of BrS patients with Nav1.5 trafficking impairment, since it has been shown to rescue the phenotype in mouse cardiomyocytes carrying a Scn5a variant [239]. Yu et al. used the heterologous systems HEK293 and tsA201 to mechanistically understand the interactions between MOG1 and Nav1.5 proteins. They proved that E83D-inducing mutation impaired the specific binding of the two proteins, consequently producing a trafficking defect of Nav1.5 channels. This translated in no effects on their transcription but caused an important I_Na_ reduction [240].

-BrS due to unknown gene mutations

As before described, BrS manifests clinically often without a clear genetic substrate although advanced technologies for genome screening are available. Without a genetic cause, familiar forms of BrS cannot be modeled with most of the in vivo, in vitro, and in silico platforms developed so far, apart from iPS derived from the patient. In 2016, Veerman et al. analyzed by patch-clamp the electrophysiological pattern of the iPS-cardiomyocytes of three different BrS patients. Surprisingly, no alterations of I_Na_, other currents, and action potential were revealed when these induced cardiomyocytes were compared to control ones. These findings led to formulating a new hypothesis in the BrS pathomechanism, for which cardiac sodium channel dysfunction is not a prerequisite for the disease onset [241].

-BrS due to drug cardiotoxicity

As in SSS, BrS might be induced by pharmacological administration of drugs with known blocking effects on I_Na_ current. Aiba et al. demonstrated as first the efficacy of some drugs for the purpose to generate an in vivo model of BrS. Right ventricular tissues from dogs were submitted to in vivo selective pharmacological perfusion with terfenadine (5 μM), pinacidil (2 μM), and pilsicainide (5 μM). High-resolution optical mapping techniques evidenced spontaneous phase-2 reentrant extrasystole, as well as polymorphic ventricular tachycardia episodes with abnormal repolarization gradient in the epicardium. Such findings lead to hypothesize a repolarization defect occurring in BrS [233].

Similarly, Morita et al. induced BrS pharmacologically in dogs with an analogous drug perfusion cocktail with the intent to evaluate whether the right ventricle was most affected due to its electrophysiological heterogenicity. They observed again a major involvement of the epicardium, while no differences in terms of action potential shape and duration were clear for the endocardium. Moreover, they detected an increased number of episodes of ventricular tachycardia in the right side of pharmacologically treated dogs [242].

More recently, Ma et al. effectively induce BrS electrophysiological impairment in iPS-cardiomyocytes derived from a healthy subject by administering in vitro ajmaline and flecainide, clinically used to unmask the ECG disease alterations and arrhythmias [234].

-BrS modeling in silico

The complex and mostly unknown genetical substrate of BrS renders particularly challenging its computational modeling and makes the attempts converging to reproduce the defective currents.

Tsumoto et al. modeled BrS human ventricle and ring computationally to examine whether local alterations of Nav1.5 channel density could create the electrophysiological substrate for phase-2 reentry. Of note, they concluded that a reduced expression of Nav1.5 channels at the lateral membrane of the cardiomyocytes induces a reduction of I_Na_ current and, hence, the loss-of-dome in the action potential, as well as phase-2 reentry [243].

Calvo et al. integrated existing mathematic models of cardiac electrophysiology to better fit the effect of neuroautonomic stimulation on the real cardiac data generated by the observation of BrS patients [244].

In silico models of BrS might be very useful to stratify the clinical risk in order to prevent fatal events in diagnosed patients. Crea et al. generated a three-dimensional model of the ST-elevated segment typically diagnosed in BrS through the right precordial leads of the ECG. The computational system confirmed the hypothesized relationship between the horizontal right ventricular outflow tract and the BrS ECG pattern [245].

A summary of the BrS modeling strategies and implications for mechanistic insights and clinical translation is provided in Table 2.

## 6. Atrial Fibrillation: Modeling a Rhythm Disturbance Highly Prevalent and Often Untreatable

Among cardiac arrhythmias, atrial fibrillation (AF) is the most common sustained CCS disturbance, accounting for about 46 million suffering subjects worldwide following a recent prevalence estimation [246]. The updated definition of AF by the ESC guidelines includes the manifestation of extra-SAN, supraventricular tachyarrhythmia, which in association to uncoordinated atrial electrical activity leads to impaired atrial contraction [247]. Unfortunately, the condition is often unresponsive to current pharmacological treatments.

AF causes can be searched among different origins: genetic mutations, hormonal dysfunction, hemodynamic instability following cardiac interventions, athletic activity in endurance, and not lastly, radiation exposure, also during war [248,249,250,251,252,253]. Social determinants, such as age (over 65), ethnicity (mainly Hispanics and Asians), and health indicators of lifestyle quality (smoking, obesity, physical inactivity, psychological stresses) are considered risk factors of AF and the disease is associated with other pathological conditions, such as heart failure, myocardial infarction, chronic kidney disease, venous thromboembolism, dementia, and cancer. Remarkably, -omics technologies have allowed researchers to better investigate the genomic, epigenomic, transcriptomic, proteomic, and metabolomic profiles of large pedigrees and the general population affected by AF [246]. As recently reviewed by Roselli et al. [254], the heritability of AF has been estimated at 62% through the genetic screenings performed with genome-wide association and sequencing studies. Several genes have been identified as mutated in AF-affected patients: among these are KCNQ1, NPPA, and TBX5.

-Modeling of AF-related KCNQ1 mutations

The KCNQ1 gene encodes the alpha subunit for voltage-gated potassium channels of I_KS_ current. Diagnosed mutations provoke a reduced atrial refractory period, due to the increased channel expression. In AF patients, variants at different penetrance were identified, as generating G229D, R231C, R231H, S209P, S140G, and V141M proteins [255,256,257,258,259].

S140G-inducing mutation was discovered as transmitted in autosomal dominant hereditariness in a 4-generation Chinese family by Chen et al. in 2003. When they expressed it in COS7 heterologous cells, patch-clamp revealed no I_KS_ current, as well as altered gating and kinetic properties (the channel was always open). Together with the alpha subunit, the minK domains codified by KCNE genes constitute the channels responsible for I_KS_ current. Only with the co-expression with KCNE1, KCNE2 or KCNE3 genes in the COS7 charging S140G protein variant, a variable increase in current density was appreciated, thus reflecting a gain of potassium channel function and an amplified outward current at depolarized potential, an easy electrical substrate for fibrillation [260].

Hong et al. diagnosed the V141M-inducing KCNQ1 mutation in a neonate presenting slow and irregular heart rate, slow ventricular response, and short QT interval at premature birth. The effects of this missense mutation were investigated in the heterologous system of *Xenopus laevis* oocytes, where KCNQ1 variant and/or KCNE1 genes were expressed. Similar to Chen et al., they concluded that V141M variant was a gain of function. Moreover, based on the data collected in *Xenopus* oocytes and available in the literature, they developed a computational model of the human heart in the context of V141M variant, which predicted the alterations in the action potential duration, but excluded modifications of the resting potential provoked by V141M mutant function [255].

Notably, the study by Campbell et al. analyzed concomitantly S140G and V141M variants in Chinese Hamster ovary (CHO) cells and revealed an enhanced sensitivity to the drug HMR-1556 when compared to wild-type channels. In addition, critical alterations were induced in I_KS_ current by S140G mutant, as larger amplitude, greater current density, shift of the voltage dependence of activation, and low deactivation time. When the S140G-inducing mutation was expressed in adult rabbit left atrial myocytes, it showed to hyperpolarize the resting membrane potential and reduce action potential duration at 90%. HRM-1556 administration (1 μM) was able to revert this trend, thus suppressing some of the AF pathologic effects on I_KS_ current [261].

-Modeling of AF-related NPPA mutations

Natriuretic peptide A-codifying gene has also been found mutated in AF, by altering the physiologic control exerted by this circulating hormone in the modulation of the neurohormonal stimulation and of ion channels [262,263,264,265]. A two-base-pair deletion was identified in the NPPA gene by Hodgson-Zingman and colleagues employing genome-wide association study and pyrosequencing. This frameshift mutation was expressed in heterozygosis in most members of the AF family. It caused the synthesis of a fusion protein including the normal hormone plus a C-terminal and induced a higher concentration in the circulation. When mutant ANP (100 nM) was circulated in an ex vivo rat heart perfusion model, a significant reduction in action potential duration at 90% and effective refractory period were observed, thus directly showing the strong implications in the shortening of atrial action potentials in AF patients carrying NPPA mutation [266].

-Modeling of AF-related mutations of TBX5 and other transcription factor genes

TBX5 is the gene found mostly mutated in genetic screenings of AF pedigrees from European and Asian countries, particularly from Iceland, Japan, and China [246]. It codifies for the homonymous transcription factor, extremely relevant for SAN developmental program. TBX5 mutations might be causative of Holt–Oram syndromic disease, which manifests with heart congenital structural defects and paroxysmal AF and is inherited as an autosomal dominant disorder [267]. Holt-Oram syndrome has been widely characterized especially for the associated defects in heart development [16,19,268,269,270]. TBX5 has also been shown to strictly regulate SCN5A expression [24].

A mouse model for Holt-Oram syndrome was proposed by Zhou et al. in 2005, in which the Tbx5 murine gene was submitted to allelic deletion. This model well recapitulated the human abnormal alterations of the inflow tract and consequent aberrant hemodynamic effects (increased peak velocity of both A and E waves) originating a deterioration of the left ventricle’s diastolic dysfunction [271]. However, no investigation was carried out on electrical function that could provide new insights on the relationship between altered hemodynamics and progressive electrophysiological impairment by the study of growing animals.

Postma et al. analyzed a three-generation pedigree, in which an atypical Holt-Oram disease manifested with few family members presenting cardiac defects, but almost all paroxysmal AF. The G125-inducing TBX5 mutation was identified as causative and studied in HEK293 cells, where it revealed normal binding to NKX2.5 but increased binding to DNA in three independent assays (EMSAs, in vitro yeast 2-hybrid X-gal assay, and pull-down assay). Moreover, the mutation exerted positive transcriptional effects on several genes, including NPPA and GJA [272].

Recently, Guzzolino et al. added a novel piece to the puzzle of Holt-Oram syndrome comprehension, by focusing on the microRNA regulating TBX5 transcriptional activity. They performed next-generation sequencing studies of the hearts isolated from wild-type, CRE, Tbx5^lox/+^, and Tbx5^del/+^ mice. These analyses evidenced an increase of miR-183 family, in particular of miR-182-5p, under control of TBX5-repressed Kruppel-like factor 4. Upon the transient overexpression of miR-182-5p in Zebrafish, profound effects on heart morphology and calcium handling could be observed. In addition, arrhythmias were evident at the ECG. A direct relationship was, thus, revealed for the first time between this miRNA and altered electrophysiological properties, by opening the way to a new possible pharmacological targeting of AF [273].

Other transcription factor genes with similar organ trophism, such as NKX2.5 and PITX2, were identified as mutated in AF. NKX2.5 mutations were found associated with AF in a genome-wide associated study of PR interval [45], in an epigenetic study on the left atrium [274], in a genetic screening of patients affected also by dilated cardiomyopathy [275], as well as diagnosed in patients with only atrioventricular conduction defects [276]. Loss-of-function was often disclosed in these studies and inherited with autosomal dominant mutations. Several animal models were generated to evaluate the effects of NK2.5 mutations causing other conduction defects than AF [276,277]. An AF-related, double NKX2.5 insufficiency modeling was realized by Chen et al. in 2019 by inducing myocardial infarction in rats in vivo and administering adenoviral vector-mediated Nkx2.5 RNA interference to HL-1 cardiomyocytes in vitro. Rats displayed a reduced level of Nkx2.5, particularly in the left atrium, after coronary artery ligation. Moreover, the RNA interference strategy was successful in decreasing NKX2.5 mRNA transcript and protein, enhancing the expression of HCN4 and almost silencing Cx40, SERCA, phospholamban, and Cav1.2, all proteins fundamental in Ca^2+^ handling [278].

Two independent studies performed in vitro and in vivo evaluated the effects of targeted mutations in cardiac transcription factor genes, as NKX2.5 and TBX5. Karakikes et al. developed through the use of TALEN technology numerous iPS cell lines, in which the genes codifying for cardiac transcription factors were knocked out. This study revealed a close control of TBX5 on extracellular matrix synthesis and identified novel targets of the transcription factor, as VCAN (versican), FN1 (fibronectin), and HSPG2 (perlecan) that, besides the known NPPA, TTN, and GJA5 (Connexin 43) genes, could be targeted for the treatment of Holt-Oram syndrome and associated AF [279]. Laforest et al. generated several models of mice haploinsufficient for Nkx2.5, Tbx5, and Gata4. The combinatorial haploinsufficiency for Tbx5 and Gata4, but not for Nkx2,5 and Gata4, was able to rescue the AF phenotype by normalizing Ca^2+^ handling. A similar effect could be obtained by reducing the expression of phospholamban [280].

-Modeling of AF-related mutations in other genes or in case of unknown mutations

Other genes were seldomly described associated to AF. An example is given by the LMNA mutations. Recent original research by Zhang et al. advanced the in vitro recapitulation of an LMNA mutation inducing familial dilated cardiomyopathy and AF. The mutation was identified by whole-exome sequencing in a Chinese male proband, who donated peripheral blood cells to generate an iPS line, to be used in the future to model LMNA-related AF [281].

Additionally, SCN5A, already described as implicated in SSS and BrS, was found mutated by Hong et al. in two AF American patients of different ethnicity, leading to two Nav1.5 protein variants (E428K and N470K). Modeling by iPS-cardiomyocytes evidenced a heart rate alteration, the prolongation of the action potential, and triggered arrhythmic events. Both variants were studied in the HEK293 heterologous system without revealing any modification in the I_Na_ current. This was, instead, significant in the AF-iPS-cardiomyocytes, in particular for the late current phase. Transcriptomic analyses and RNA sequencing in AF-iPS-cardiomyocytes revealed the dysregulation of a consistent number of genes by the two SCN5A mutations. The nitric oxide (NO) signaling resulted to be significantly impaired. The administration of the NO blocker N-ethylmaleimide was, in fact, able to reduce the late I_Na_ current. More interestingly from a translational point of view, the drug ranolazine (20 µM) rescued the aberrant arrhythmogenic behavior observed in these AF-iPS-cardiomyocytes [152], suggesting a possible, personalized pharmacological treatment for these AF patients.

The genetic etiology of some AF forms can be very complex or remains completely unknown although in-depth screenings have been performed. By means of whole-exome sequencing, Benzoni et al. disclosed that no unique gene mutations could be identified in three AF patients non-responsive to pharmacological treatment. Notably, one hundred genetic variants were shared by these three subjects, revealing an extremely complex genetic background. The patient-specific iPS lines differentiated in cardiomyocytes similar to control ones but showed a higher beating rate and longer action potential. In these cells, I_Ca,L_ was increased, but no changes in potassium currents were observed, apart from I_KS_ and I_K1_ that were not established after iPS differentiation. Delayed after-depolarizations and ectopic beats could be registered after isoproterenol and E4031 challenge [282]. Chiu et al. investigated the genetic cause(s) of AF in three patients before generating their iPS from the peripheral blood mononuclear cells for next mechanistic studies. Mutation analyses were carried out applying different protocols but no apparent gene mutation was identified [283].

-Other AF modeling modalities

Neurostimulation protocols have been adopted quite early in CCS research to create AF models in vivo. A rapid atrial pacing has been demonstrated to effectively generate an AF classical pattern. Fareh et al. submitted dogs to a 400-bpm neurostimulation for 7 days, who ultimately showed a decrease in the atrial effective refractory period, but an increase in its heterogenicity, as well as duration and the propension to AF episodes. The drug mibefradil rescued the AF phenotype in a dosage-dependent manner [284].

Denham et al. used a similar approach to induce AF in sheep. The neurostimulation was carried out by the application of a neurostimulator delivering 50 Hz pacing on a 30 s on/off cycle for 8 weeks. Since sustained AF developed in the animals displaying higher fibrillation numbers before pacing, a direct relationship between the two parameters was hypothesized. It was, hence, suggested to use fibrillation number as a predictor of sustained AF [285].

As previously mentioned, animal AF modeling can be achieved through experimental acute myocardial infarction by coronary artery ligation and/or consequent heart failure. The most critical pathologic event underneath heart dysfunction associates with an impairment in Ca^2+^ handling, in particular, RYR2 dysregulated functionality. After coronary artery ligation, Nofi et al. [286] electrically stimulated infarcted rat hearts in vivo with 200-bpm burst pacing at 50 Hz, successfully generating AF. In order to stabilize the arrhythmogenic hearts, dantrolene was administered, proving once again its efficacy in suppressing triggered arrhythmias in atrial and ventricular cardiomyocytes [287,288].

-Modeling AF through in silico reconstruction

A recent review by Heijman et al. described the milestones in the computational modeling of AF starting from 1952 with the Hodgkin–Huxel model and reaching 2020 with the Hansen study [289]. The increasing understanding and novel insights gained by all studies in atrial physiology and dysfunction were progressively joined together. Nowadays, these systems offer incredible detail in subcellular, cellular and tissue/organ-level simulation, allowing to generate accurate predictions relevant also for clinical risk stratification. Current in silico AF modeling exploits integrated bioinformatic analyses, deep, and machine learning also to select and predict the effect of a drug therapy in a patient following a pathomechanism-based precision medicine approach.

Belletti et al. proposed an AF computational model studying the functionality of several mutated potassium channel proteins, as KCNE3 displaying V17M modification, KCNH2 with T895M or T436M. All these variants induced a gain-of-function of the K channels with consequent higher current density, reduction in the duration of the action potential, and leveling of the restitution curves. The model predicted that the worse effects were generated by V17M-KCNE3 channel [290].

Diagnostic predictive models were developed by several groups with the intent to stratify the risk for AF development in patients affected for example by hypertension, diabetes, or hypertrophic cardiomyopathy [291,292], by obtaining further indications on the association between the most severe forms of the latter and AF. Deep learning was shown in an experimental paced canine model to predict AF progression based on conduction heterogenicity, action potential characteristics, and other rhythm variants possibly associated with other concomitant CVD diseases [293]. Taking into consideration that AF can manifest as secondary to structural defects/variants, also induced by cardiac surgery, Duenas-Pamplona et al. developed a computational fluid dynamics simulation of the patient-specific left atrium, in which several predicting indexes were proposed, including thrombosis-mediated stasis evaluation [294].

One of the most intriguing applications of in silico AF modeling is definitely related to precision medicine pharmacology. Several AF forms are still insensitive to current drug therapy and calibrated models could help to develop more efficacious treatments. As an example, Bai et al. reported the development of population-based simulations in AF due to a transcription factor gene mutation. The electrical remodeling caused by the mutation on the action potential was recapitulated. The assessment of class I antiarrhythmic drugs disopyramide, quinidine, and propafenone in this simulation system allowed us to establish that the first drug could more successfully treat this familiar AF [295].

A summary of the AF modeling strategies and implications for mechanistic insights and clinical translation is provided in Table 3.

## 7. Conclusions

From the first simplistic experimental assays to the current comprehensive, sophisticated simulations, in vitro, in vivo, and in silico modeling has persistently confirmed its relevance to enhance the fundamental knowledge and gain new insights on the physiology and pathophysiology of body functions, CCS included. Rhythm diseases are extremely heterogenous, and, as shown by the representative disturbances overviewed in this review, their pathomechanism might be complex and often not understood yet. In addition, patients affected by a specific rhythm disturbance might show differential responsiveness to current pharmacological treatments, thus pointing out the need for more specific approaches.

Animal models, especially mammalian ones, have offered the possibility to observe the CCS and its disorders in a living, open system, often humanized by transgenesis. However, many differences exist in terms of anatomy and electrophysiology with humans and might render it difficult to fully recapitulate human arrhythmias.

Cellular models are easier to handle than animals and possibly more physiologic, especially when isolated from humans. The unavailability of human primary cultures of CCS due to cell instability after isolation has represented an important bottleneck that was overcome by the advent of iPS technology. By the application of more and more advanced protocols, iPS have been differentiated to nodal, atrial, and ventricular cardiomyocytes, thus providing the unprecedented opportunity to study the CCS diseases in the Petri dish. Relative immaturity of iPS-cardiomyocytes in terms of current density and consequent altered resting membrane potential might, however, underrepresent the real sub/cellular signs of the rhythm disturbance and new strategies must be studied to push further their differentiation.

Although not tissue- and organ-specific, heterologous cellular systems, as HEK293, CHO, COS7, and tsA201 cells, are still currently used in tandem with in vivo models and/or in vitro iPS-cardiomyocytes to verify trafficking, assembly, sensory, and/or gating alterations that could be provoked by gene mutations. In addition, they have allowed us to establish whether a particular gene variant could induce a gain- or loss-of-function for the mutated channel protein and the consequent effect on the current homeostasis. Nevertheless, heterologous systems with their unique expression of the mutant and/or wild-type proteins might fail to reproduce the peculiar current impairment, when the genetic background is complex, or the genetic causes might be more than the one known.

The advancement in the field of in silico CCS modeling is astonishing and parallels the availability of new imaging, -omics, bioengineering, and medical technologies, as well as novel clinical insights generated. Computational models gather together all the information gained on a given rhythm disturbance with the intent of effectively modeling sub/cellular, tissue, and organ variations, predicting the effects of a certain mutation or system perturbation, obtaining a risk stratification, and, not lastly, developing personalized medicine approach for affected patients. In this scenario, model validation is thus fundamental in order to increase the refinement and enhance prediction power, as shown by the numerous cardiac rhythm-focused models developed in around 70 years.

Although the way is still long towards the full comprehension of SSS, BrS, AF, and other CCS rhythm disturbances, recent progress in their in vitro, in vivo, and in silico modeling render closer the identification of underlying pathomechanisms and the development of efficacious treatments tailored for each specific patient.

## Figures and Tables

**Figure 1 cells-10-03175-f001:**
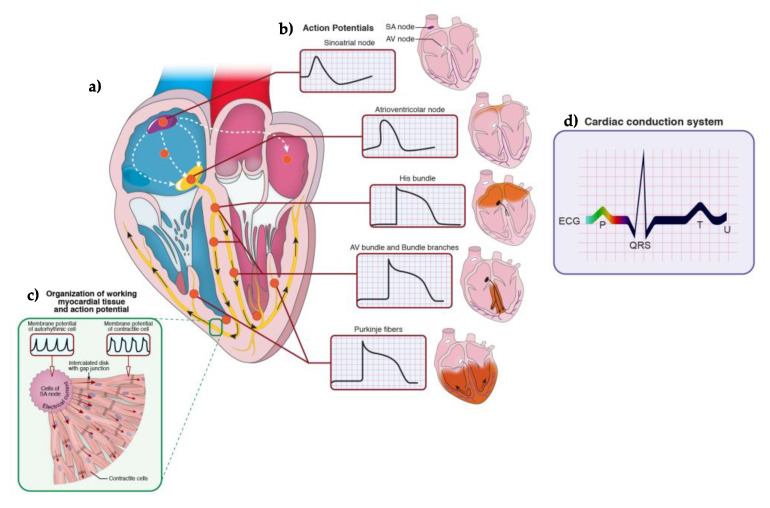
The cardiac conduction system (CCS). (**a**) The stations of the CCS and their typical action potential: sinoatrial node (SAN), atrioventricular node (AVN), His bundle, AV bundle and bundle branches, and, finally, Purkinje fibers; (**b**) Heart regions activated by the CCS electrophysiological activity during the cardiac cycle; (**c**) The working myocardium: architecture and action potential; and (**d**) Electrocardiogram (ECG) of a healthy subject.

**Table 1 cells-10-03175-t001:** Sick Sinus Syndrome (SSS): causes, modeling strategies, and their clinical impact.

Disease Causes	Modeling Strategies	Disease Recapitulation Pattern	Novel Mechanistic Insights	Clinical Translation Potential	Ref.
**Genetic mutations**					
SCN5A	In vivo model with Scn5a^+/−^ mice	SAN bradycardia, conduction, and exit block, typical of SSS, are recapitulated.	Role of SAN I_Na_ current on atrial activation		[161]
		Prolonged RR intervals (ventricular repolarization)	Sex- and age-dependent effects caused by the genetic variant		[162]
		Electrophysiological alterations worsened in tandem with fibrosis	Up-regulation of TGF-beta1 induced colagen synthesis increase		[163]
	In vivo, in vitro and in silico model based on Scn5a^+/−^ rabbit	Characterization of post-natal SAN I_Na_ current in vivo; SAN cells displayed in vitro different pacing abilities based on their localization, with a dramatic decrease in the periphery	In silico modeling reconstructed I_Na_ current modifications related to cell localization, sex and aging.		[164]
HCN1	In vivo and in vitro HCN1^−/−^ mouse model	In vivo: prolonged SAN recovery time, RR interval, high beat-to-beat dispersion, and sinus pauses. In vitro: no alteration in the expression of other ion channels and proteins, but decreased beating rate, abnormal beat-to-beat interval, and delayed impulse formation	No established compensation revealed the HCN1 relevance in the generation of a basal depolarizing current.	HCN1 mutations have never been identified in humans, thus these findings are not completely translatable.	[166]
HCN4	In vitro HCN4^mut^ heterologous system	A rabbit HCN4 mutated gene was transfected in COS-7 cells. The D553N mutated channel displayed a modification in a region connecting two domains. Faster activation and slower deactivation were observed in transfected COS-7 cells when compared to their control.	The mutation did not affect channel voltage-dependence.	This mutation is similar to the one reported for a SSS patient, responsible for the generation of sinus bradycardia, cardiac arrest, polymorphic ventricular tachycardia, and torsade de pointes), thus personalized treatments could be developed targeting the connecting region of the mutated channel.	[167]
	In vitro HCN4^mut^ heterologous system and in silico modeling	Human HCN4 missense mutations of two young SSS patients were expressed in *Xenopus laevis* oocytes and COS-7. A complete loss of function was induced in homo- or heteromeric combinations due to a reduced HCN4 membrane expression. Collected data from control and transfected cells were used to develop a computational model.	A new hypothesis was formulated on the possible structural channel alteration interfering with trafficking and voltage sensing.	Personalized pharmacology approaches for SSS patients bearing these HCN4 mutations could be designed on this new mechanistic hypothesis.	[169]
NCX	In vivo modeling through NCX^−/−^ mice	Intracellular Na^+^ alterations due to Na^+^/Ca^2+^ exchanger variant protein were studied and revealed to be typical of SSS phenotype.	These pathological modifications were correlated to alterations in the feedback between Ca^2+^ handling and membrane potential clock.		[170]
	In silico modeling through NCX^−/−^ mice				[171]
DSP	In vitro modeling with iPS-cardiomyocytes of a DSP^mut^ patient	Impaired function of Nav1.5 and L-type Ca^2+^ channels was revealed in these cells bearing a H1684R-inducing mutation.	Further studies with other DSP^mut^ iPS are needed to confirm these alterations of the electrical phenotype, as typical manifestations of SAN dysfunction in the mutation context.		[176]
**Aging**					
	In silico modeling of rat SAN aging	The action potential of a SAN cell derived from an aging rat was reconstructed computationally through literature studies.	SAN dysfunction was demonstrated to be originated by concomitant, age-dependent alterations in several currents and electrical functions.		[178]
**Inflammation**					
G-protein coupled receptors in the context of catecholamines and angiotensin II storm during cardiac injury	In vitro and in vivo modeling with ES- derived cardiomycytes and knock-in mice	Suppression of binding and inhibition of G-protein coupled receptors with their negative modulators, i.e., regulators of G protein signaling (RGS) was studied in vitro and in vivo in the settings of isoproterenol, adenosine, and muscarinic M2 receptor agonist administration to simulate SAN automaticity during inflammation.	Catecholaminergic stimulation did not have effect, while adenosine and muscarinic M2 receptor agonist induced SAN bradycardia in RGS-insensitive mutant ES-cardiomyocytes and in knocked-in mice. These findings revealed the ability of RGS to control SAN automaticity independently from neurohormonal stimulation.		[191,192,193]
Mitochondrial oxidative stress	In vivo modeling with thioredoxin 2^−/−^ mice	Thioredoxin 2 deletion in whole mouse hearts induced dilated cardiomyopathy, AV block, and sinus bradycardia, following HCN4 downregulation.	HDAC4-MEF2C signaling pathway modifications were reported as the mechanistic link between mitochondrial oxidative stress and reduced HCN4 expression in SAN cells.		[194]
Myocardial ischemia/reperfusion injury	In vivo modeling by experimental clamping of SAN region in rats	The clamping of rat SAN region with hemostatic forceps induced injury in terms of anatomical cell distribution. A reduction in HCN4 and SCN5A expression and Ca^2+^ handling concurred to generate decreased heart rate and long RR intervals.	The herbal pharmacological compound Zengl Fumai granule was administered by acting on pathways related to myocardial ischemia/reperfusion injury, as TRIM genes.	Zengl Fumai granules were shown to rescue SSS phenotype in vivo. Its action was confirmed in vitro with human AC16 cardiomyocyte line. Further experiments are needed to demonstrate whether it could have a similar effect in the clinical pharmacology treatment of SSS.	[195,196,197]
	In vitro ischemia/reperfusion injury model with rabbit SAN cells	A remarkable reduction of the I_f_ current was observed in rabbit SAN cells after ischemia/reperfusion injury.	The herbal medicine astragaloside increased HCN4 expression and protected cells from stress responses.	Astragaloside ability to suppress SSS signs needs to be verified in other experimental settings and in the clinic.	[198]
	In silico multiscale simulation of SAN-atrium dysfunction	Experimental observations on ion current modifications in rabbit SAN cells after ischemia/reperfusion injury were used to develop a computational model. I_NaCa_ and I_K_ were impaired in ischemic settings. Additionally, SAN heart rate and atrial conduction velocity were decreased.	Simulated acetylcholine vagal stimulation worsened both ischemic consequences and SAN dysfunction, leading to arrest and exit block.		[199]
**Drug or chemical cardiotoxicity**					
Streptozotocin	In vivo rodent model	Both hyperglycemia and streptozotocin (used to induce diabetes) induced SSS in rats. Streprozotocin provoked systemic inflammation, high ROS levels, cardiac fibroblast apoptosis, decreased HCN4 expression, increased TGF-beta1 expression, fibrosis, macrophage infiltration and reduced baroflex sensitivity.		The complex pathological phenotype of diabetes and SSS could be reversed in vivo by administering IL-10. Further studies are needed to confirm the ability of IL-10 in suppressing SSS signs related to diabetes.	[203]
Angiotensin II	In vivo and/or in vitro cardiac overload	Angiotensin II/Smad pathway activation induced SAN fibrotic lesions.	The transient receptor potential subfamily M member 7 (TRPM7) was found implicated in cardiac fibrosis.	TRPM7 could be investigated as a target to prevent SAN fibrosis in SSS.	[204]
		Angiotensin II continuous administration in rats provoked typical SSS signs.	In HL-1 atrial cardiomyocytes (used as SAN cells surrogate) in vitro, angiotensin II mediated PKC/NOX-2 signaling upregulation, as well as reduced HCN4 and HDAC4 levels.	Shenxian-Shengmai herbal medicine generated protection from oxidative stress both in vitro and in vitro. The ability of this herbal medicine to contrast SSS signs due to angiotensin II overload needs to be confirmed.	[205]
Pinpoint press permeation	In vivo models	Pinpoint permeation with 10% sodium hydroxide reduced SAN automaticity less than surgical clamping, but equal, prolonged times of sinoatrial recovery and conduction.		Useful artificial models to reproduce SSS phenotype, although not physiological.	[195]
		Local administration of 20% formaldehyde provoked atrial remodeling with inflammation and alteration of extracellular matrix homeostasis.			[206]

Legend: SSS: sick sinus syndrome; SAN: sinoatrial node; RGS: regulators of G protein signaling; TRPM7: transient receptor potential subfamily M member 7.

**Table 2 cells-10-03175-t002:** Brugada syndrome (BrS): causes, modeling strategies, and their clinical impact.

Disease Causes	Modeling Strategies	Disease Recapitulation Pattern	Novel Mechanistic Insights	Clinical Translation Potential	Ref.
**Genetic mutations**					
SCN5A haploinsufficiency by targeted gene disruption	In vivo, in vitro, and/or ex vivo models based on Scn5a^+/−^ mice	Scn5a^+/−^ mice showed normal QT interval duration in respect to wild-type animals, but conduction block, re-entrant arrhythmias, and ventricular tachycardia were observed. Their ventricular cardiomyocytes display a 50% reduction of INa current without gating properties changes, slowed conduction velocity.	Scn5a^−/−^ mice could survive only in uterus until gestational day E10.5 and manifested ventricular morphogenetic defects.	Scn5a^+/−^ mice generated by gene targeted disruption have BrS signs similar to humans but do not bear the same genetic mutation.	[214]
		QRS interval prolongation degree was used to classify severity, which was more important in old Scn5a^+/−^ mice observed in vivo and ex vivo.	A correlation between severity and INa current reduction was observed, as associated to normal mRNA levels but decreased protein product.	QRS interval prolongation degree is more effective than aging as severity marker. Scn5a^+/−^ mice generated by gene targeted disruption have BrS signs similar to humans but do not bear the same genetic mutation.	[215]
		In Langendorff ex vivo system, maximum conduction velocity was higher in the endocardial surface than in the epicardial one of Scn5a^+/−^ right ventricles. Fleicanamide and quinidine worsened this conduction block.		Wavelength restitution was found to be a better prognostic marker of action potential duration. More effective prediction could be obtained by alternans magnitude, instead of considering the sole diastolic interval. Scn5a^+/−^ mice generated by gene targeted disruption have BrS signs similar to humans but do not bear the same genetic mutation.	[216]
		Scn5a^+/−^ mice showed a cleared transmural gradient in action potential duration between the right and left ventricles differently from wild-type animals. By increasing pacing, pacing delays were observed only in the right ventricles of mutants.		The physiological anatomical dissimilarities between the right and left ventricles contribute to create a structural, electrophysiological substrate for BrS manifestations if Nav1.5 channels are mutant. Scn5a^+/−^ mice generated by gene targeted disruption have BrS signs similar to humans but do not bear the same genetic mutation.	[217]
		Ex vivo, hearts from old Scn5a^+/−^ mice and cardiac tissue preparations from young animals were stimulated with carbachol and isoprenaline. Longer effective refractory periods were observed for Scn5a^+/−^ cardiac organs and tissues.		These results suggest a loss of the ability to reply to autonomic stimulation in case of Scn5a haploinsufficiency. Scn5a^+/−^ mice generated by gene targeted disruption have BrS signs similar to humans but do not bear the same genetic mutation.	[218]
		In the context of Scn5a haploinsufficiency, a reduction of I_Na_ and I_K1_ was observed, which were due to changes in Nav1.5 and Kir 2.X mutants		Kir2.1 overexpression in Scn5a^+/−^ mice increased I_K1_ and I_Na_, too, with rescuing effect on BrS pathologic phenotype. Further studies need to be performed to evaluate whether a Kir2.1 overexpression gene therapy could be effective in the treatment of human BrS. Scn5a^+/−^ mice generated by gene targeted disruption have BrS signs similar to humans but do not bear the same genetic mutation.	[220]
SCN5A point mutations	In vivo and in vitro mouse model based on human 1798insD mutation	Mice bearing 1798insD mutation in homozygosis did not survive. Heterozygous mice showed prolonged PQ, QRS, and QTc intervals. Flecainide exacerbated these alterations, causing sinus bradycardia and/or arrest. In addition, total activation time and effective refractory periods were found different in right and left ventricles. In vitro, cardiomyocytes isolated from these animals displayed longer action potential and reduced upstroke velocity. This model reproduced mixed SSS-BrS phenotype (predominant RV affection, SAN dysfunction, and absent ventricular arrhythmias), already observed in the Dutch family harboring 1798insD mutation.	This mouse model could provide further insights on the altered molecular pathways and identify targets for a personalized pharmacological approach for this cardiopatic patients.		[222]
	In vitro heterologous system to express L325R variant-inducing mutation and in silico model	1:1 overexpression of wild-type and L325R variant-inducing SCN5A mutation in HEK-293 cells evidenced a decrease in Nav1.5 channels and derived current. Computational simulations demonstrated premature repolarization and dome loss in the action potential of cardiomyocytes bearing this protein variant.		The computational model allowed to observe a correlation between I_Na_ current reduction and fever, clinically observed in patients bearing this SCN5A mutation. For the first time, the concept of dominant-negative effect was introduced in BrS diagnosis.	[223]
	In vitro heterologous system, cardiomyocyte overexpression and in vivo mouse modeling of R104W, R121W, and/or Y87C mutants	The unique overexpression of R104W variant in HEK-293 and rat neonatal cardiomyocytes led to a complete I_Na_ suppression. 1:1 overexpression with wild-type gene induced a less critical I_Na_ reduction. Mice bearing this mutation manifested lower heart rate, I_Na_ reduction, prolonged RR intervals, longer P-wave.	R104W mutant was not able to reach the cell membrane, due to trafficking impairment. Through these different models, it was possible to confirm the dominant-negative effect of this mutation, related to a defective N-terminal.		[224,225]
		Several N-terminal variants, including R104W and R121W, and the mutated protein Y87C were overexpressed in human kidney TsA-201 and COS-7 cell lines. Mutant proteins showed binding to calmodulin (weak for R121W variant) and wild-type Nav1.5 channels.	These experiments allowed to confirm a dominant-negative effect for both R104W and Y87C variants (not for R121W). Moreover, they identified the existence of a binding site for calmodulin in the N-terminal of Nav1.5 protein. These novel mechanistic insights need to be confirmed in a more physiological model.		[226]
	In vivo modeling through E558X-expressing pigs	This human mutation was overexpressed in Yucatan minipigs. No events of sudden death were reported in two-year follow up, but animals had slow conduction, prolonged P wave, PR, and QRS intervals, sustained atrial-His and His-ventricular conduction periods. SAN recovery time was prolonged and conduction velocity decreased. Nav1.5 protein was decreased confirming electrophysiological observation of reduced I_Na_ current. Aging and temperature rise aggravated these signs, however without any difference in between right and left ventricles.		This model did not recapitulate other human BrS signs, as myocyte hypertrophy or fibrosis.	[227]
	In vitro modeling with iPS-cardiomyocytes of a 1795insD SCN5A patient	Reduced I_Na_ current, decreased upstroke velocity and prolonged action potential were observed in patient-specific iPS-cardiomyocytes and were worsened by increasing pacing.	These findings were similar to the ones revealed in mice carrying the same mutation and their derived iPS.	The late sodium current inhibitor GS-458967 was able to rescue the BrS phenotype in both mouse and human Brs iPS cells. Electrophysiological immaturity of iPS-cardiomyocytes (absent I_K1_ current) renders unfeasible to generate phase-1 repolarization and hence to effectively simulate all BrS signs.	[229,230]
	In vitro modeling with patient iPS-cardiomyocytes expressing SCN5A variants (R620H/R811H and prematurely truncated channel)	For all variants, the action potential was abnormally prolonged, with enlarged variability of the peak-to-peak interval and slower depolarization. I_Na_ current was decreased, due to a reduced number of Nav1.5 channels. Ca^2+^ handling was altered with reduced transient amplitude and maximal intracellular concentration.	Downregulation of Ca^2+^ handling paralleled cardiac hypertrophy and beta-adrenergic stimulation.	Targeted genome editing by CRISPR-CAS9 technology could be used to rescue the phenotype by mutation correction. Electrophysiological immaturity of iPS-cardiomyocytes (absent I_K1_ current) renders unfeasible to generate phase-1 repolarization and hence to effectively simulate all BrS signs.	[231]
	In vitro modeling with patient iPS-cardiomyocytes expressing R367H-inducing SCN5A variant and comparison with heterologous system	BrS iPS-cardiomyocytes did not show morphological alterations, but electrophysiological changes, as I_Na_ current reduction (30%), increased activation, decreased steady-state deactivation, and accelerated recovery from inactivation.	Heterologous system based on tsa201 cells revealed the causal role in defective functionality of the mutant channels. iPS-cardiomyocytes possess however a stronger potential to mimic BrS.	Electrophysiological immaturity of iPS-cardiomyocytes (absent I_K1_ current) renders unfeasible to generate phase-1 repolarization and hence to effectively simulate all BrS signs.	[232]
	In vitro modeling with patient iPS-cardiomyocytes expressing compound A226V and R1629X-inducing SCN5A variant and comparison with heterologous system	R1629X variant expressed in tsA201 cells induced complete I_Na_ abolishment due to trafficking impairment, while A226V variant reduced by half I_Na_ current due to halved SCN5A transcript. iPS-cardiomyocytes displaying A226V variant did show current reduction but not so importantly and differ from the controls in terms of recovery rate from inactivation. The missing IK1 current was introduced, so that a dome-like pattern could be visible during phase-1 repolarization. By decreasing pacing and increasing temperature, the electrophysiological pattern was aggravated.	The introduction of I_K1_ current in iPS-cardiomyocytes allows to phenocopy the typical BrS action potential and thus, represents a more physiological disease model, in which to test possible pharmacological treatments.		[234]
	In vitro modeling with healthy iPS-cardiomyocytes induced to express A735V SCN5A variant through CRISPR-CAS9 and comparison with heterologous system	Although Nav1.5 transcripts were equally expressed with respect to controls, a decreased upstroke velocity and a positively shifted curve of voltage-dependent activation could be appreciated. The use of the heterologous system confirmed the results observed in engineered iPS-cardiomyocytes.	Trafficking impairment was found not to be the reason for observed pattern.	Hybrid modeling demonstrated to be a valid experimental choice to gain more information on the disease.	[236]
Other mutated genes	In vitro modeling with patient iPS-cardiomyocytes expressing compound c.3803G>A and c.3749G<A SCN10A variant	In iPS-cardiomyocytes, SCN5A and SCN10A transcripts resulted to be increasingly expressed. Peak I_Na_ current, _ICa,L_, and I_NaCa_ were reduced, as well as the action potential amplitude and maximum depolarization velocity, while delayed afterdepolarizations and ectopic beats became more frequent events.			[237]
	In vitro heterologous MOG1 expression systems	E83D-inducing MOG1 mutation was shown to affect Nav1.5 trafficking in HEK293 and tsA201.	E83D-inducing mutation was demonstrated to compromise the binding of MOG1 protein with Nav1.5 channels.	MOG1 overexpression could be to induce a rescue, even in case of SCN5A mutation	[239,240]
Unknown gene mutations	In vivo modeling with patient iPS-cardiomyocytes without any known genetic mutations	iPS-cardiomyocytes derived from three patients with unknown mutations after genomic screening did not display INa or other current impairment.	These observations led to formulate a new mechanistic hypothesis, for which Nav1.5 channel dysfunction is not a prerequisite for BrS onset.		[241]
**Drug cardiotoxicity**					
Terfenadine, pinacidil, and pilsicainide	Ex vivo model of drug cardiotoxicity	Ex vivo perfusion of canine right ventricular tissue with a cocktail of terfenadine, pinacidil, and pilsicainide induced spontaneous phase-2 reentrant extrasystole, polymorphic ventricular tachycardia, and abnormal epicardial repolarization gradient.			[233]
	In vivo model of drug cardiotoxicity	When the cocktail was administered to dogs, ventricular tachycardia manifested predominantly in right ventricles by interesting the epicardium, without alterations in the endocardium.			[242]
Ajmaline and flecainide	In vitro iPS-cardiomyocyte model of drug cardiotoxicity	Ajmaline and flecainide are used to unmask the ECG BrS signs. When administered to iPS-cardiomyocytes from a healthy donor, they provoked the classical BrS pattern.			[234]
**In silico modeling**					
	In silico modeling of BrS human ventricle	I_Na_ alterations were reproduced in silico.	A reduced Nav1.5 channel expression was found to impair I_Na_ current and induce the loss-of-dome and phase-2 reentry.		[243]
	In silico modeling of ST-elevated segment	The typical ST-elevated segment registered at the precordial leads was reproduced.		This model allowed to confirm the hypothesized relationship between horizontal right ventricular outflow tract and BrS pattern. In silico models can be useful in stratifying the clinical risk.	[245]

Legend: BrS: Brugada syndrome; SSS: Sick sinus syndrome; SAN: sinoatrial node; iPS: induced pluripotent stem cells.

**Table 3 cells-10-03175-t003:** Atrial fibrillation (AF): causes, modeling strategies, and their clinical impact.

Disease Causes	Modeling Strategies	Disease Recapitulation Pattern	Novel Mechanistic Insights	Clinical Translation Potential	Ref.
**Genetic mutations**					
KCNQ1	In vitro heterologous systems and/or computational models	S140G-inducing KCNQ1 mutation, discovered in a 4-generation Chinese family with AF, was expressed in COS7 cells. No IKs current and altered gating and kinetic properties were observed.	Co-expression of S140G variant and several KCNE genes (normally not expressed in COS7 cells) revealed a gain of potassium channel function and increased outward current at depolarized potential, thus evidencing the electrical substrate leading to AF.		[260]
		The V141M-inducing missense mutation was expressed in Xenopus laevis oocytes. A computational model simulating human heart behavior in the context of V141M variant.	This mutation was found to induce a gain of function. Through the computational model, alterations of the action potential duration in human ventricular cardiomyocytes and sinoatrial node cells could be predicted, by excluding modifications of the resting membrane potential.		[255]
		Compound S140G- and V141M-inducing KCNQ1 mutation, observed in Chinese AF patients, was expressed in CHO cells and adult rabbit left atrial myocytes leading to several electrophysiological signs typical of AF.	The drug HRM-1556 was able to revert AF signs when tested both on CHO cells and rabbit atrial cardiomyocytes. Further tests need to be performed in a more physiological model.		[261]
NPPA	Ex vivo rat model	A mutant ANP resulting from a two-base-pair deletion observed in AF patients, was circulated in an ex vivo rat heart perfusion model. Treated hearts manifested a significant reduction in action potential duration at 90% and effective refractory period.		The strong implications in the shortening of atrial action potentials in AF patients with this NPPA mutation were revealed.	[266]
Transcription factor genes	In vivo modeling with Tbx5^+/del^ mice	Holt Oram syndrome, related to TBX5 mutation expressed in heterozygosis in European and Asiatic AF patients, was modeled in mice by allelic deletion of Tbx5, by recapitulating typical inflow tract alterations, consequent hemodynamic alterations in A and E waves, and left ventricle diastolic dysfunction.	No investigation on electrical function was performed, thus no novel insights on the relationship between altered hemodynamics and electrophysiological impairment were achieved.		[271]
	In vitro heterologous system of TBX5 mutation	G125-inducing TBX5 mutation was expressed in HEK293 cells. The variant bound normally to NKX2.5 but increased to other DNA fragments.	The mutation was revealed to exert a positive effect on the transcription of several genes, as NPPA and GJA.		[272]
	In vivo overexpression of TBX5-modulating miRNA in Zebrafish model	miR-182-5p, found increased in Tbx5^+/del^ mice, was overexpressed in Zebrafish, leading to profound effects in heart morphology and calcium handling, as well as heart arrhythmic behavior.	These findings related to miR-183 family overexpression could open up to new pharmacological treatments targeting miR-182-5p.		[273]
	In vivo and in vitro modeling simulating NKX2.5 mutations	A myocardial infarction-heart failure rat model was developed to reproduce AF in vivo. Nkx2.5 was found to be reduced in its transcription, particularly in the left atrium. In HL-1 cardiomyocytes, RNA interference decreased both NKX2.5 transcripts and protein.	The in vitro model revealed that NKX2.5 downregulation induced increased HCN4 expression, while importantly reduced the expression of proteins involved in Ca^2+^ handling.		[278]
	In vitro modeling with iPS-cardiomyocytes by knocking out of transcription factor genes	TALEN technology was used to induce knock out of transcription factor genes in iPS and differentiated cardiomyocytes were studied, revealing a fundamental role of TBX5 on extracellular matrix synthesis.	Novel targets of TBX5 were identified: VCAN, FN1, and HSPG2 could be targeted for a pharmacological treatment of Holt Oran syndrome.		[279]
	In vivo modeling with Nkx2.5, Tbx5, and Gata4 haploinsufficient mice	Several combinatorial haploinsufficient mice were generated.	Only the combinatorial haploinsufficiency for Tbx5 and Gata4 was able to rescue AF phenotype by normalizing Ca^2+^ handling. Further tests should be performed to confirm these findings in a more physiological system.		[280]
Other genes or unknown genetic causes	In vitro modeling with LMNA-mutated iPS-cardiomyocytes	Peripheral blood of a Chinese AF patient carrying a LMNA mutation was used to generate an iPS line.	This iPS line could be employed in an in vitro model shedding more light on the AF pattern in the context of LMNA mutations.		[281]
	In vitro modeling with patient iPS-cardiomyocytes expressing compound E428K- and N470K-inducing SCN5A variant and comparison with heterologous system	These SCN5S mutations were responsible for the AF pattern of an American patient. Patient-specific iPS-cardiomyocytes evidenced heart rate alteration, action potential prolongation, and triggered arrhythmias. Variants studied in HEK293 cells did not reveal I_Na_ current modifications, which were instead significant in iPS-cardiomyocytes.	Gene expression analyses and RNA sequencing of iPS-cardiomyocytes revealed dysregulation of several genes and alteration of nitric oxide signaling.	The administration of the nitric oxide blocker N-ethylmaleimide reduced the late I_Na_ current. In addition, ranolazine rescue the arrhythmic behavior of AF iPS-cardiomyocytes.	[152]
	In vitro modeling with patient iPS-cardiomyocytes with unknown genetic etiology	No unique gene mutation could be identified in three AF patients unresponsive to pharmacological treatment. Derived iPS were differentiated to cardiomyocytes and used to study the AF pattern in vitro. Higher beating rate, longer action potential, and increased I_Ca,L_ were observed. Isoproterenol administration provoked delayed after-depolarizations and ectopic beats.	This model could be employed to develop effective pharmacological treatments so far unavailable for these three AF patients.		[282]
**Experimental induction**					
Neurostimulation	In vivo modeling with dogs	A 400 bpm-neurostimulation of dog hearts for 7 days induced AF signs.	The administration of the drug mibefradil rescued the diseased phenotype.		[284]
	In vivo modeling with sheep	A 50 Hz pacing of sheep hearts for 8 days induced AF signs.	Statistical analyses performed on lectrophysiological observations led to identify the fibrillation number as a predictor of sustained AF.		[285]
Myocardial infarction-heart failure- induced AF	In vivo modeling by coronary artery ligation and neurostimulation in rats	Rat hearts were submitted to coronary artery ligation and 200 bpm-neurostimulation, by successfully inducing AF onset.	Dantrolene suppressed triggered arrhythmias.		[286]
**In silico modeling**					
	In silico modeling of V17M-KCNE3, T895M-KCNH2, and T436M-KCNH2	Computational simulations revealed that all these variants induce a gain-of-function of the K channels.	This model was able to predict that V17M-KCNE3 variant was responsible for the worse AF scenario, and hence might be relevant for risk stratification.		[290]
	Deep learning from an experimentally paced canine model	Observations collected from a neurostimulated canine heart model were submitted to deep learning.	Based on conduction heterogenicity, action potential characteristics, other rhythm modifications, and concomitant cardiovascular diseases, AF progression can be predicted.		[293]
	Computational model of AF left atrium fluid dynamics	Modifications of fluid dynamics, also induced by surgery, were modeled in silico.	Such a modeling system allows to identify predicting indexes for AF progression and severity, as thrombosis-mediated stasis.		[294]
	Population-based computational simulations	Observations collected from a large cohort of patients with the same transcription factor gene mutation were loaded in a computational model able to reproduce the electrical remodeling and where several drugs, as disopyramide, quinidine, and propafenone, could be assessed to predict their potential efficacy.	This in silico model allowed to establish that only disopyramide could be effective in the treatment of this AF genetic form. Thus, similar modeling approaches have a strong potential for the advancement of personalized medicine treatments.		[295]

Legend: AF: Atrial fibrillation; iPS: induced pluripotent stem cells.

## Data Availability

Not applicable.

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
