# Peer review of "Inherited and Acquired Rhythm Disturbances in Sick Sinus Syndrome, Brugada Syndrome, and Atrial Fibrillation: Lessons from Preclinical Modeling"

_cells, 2021, doi:10.3390/cells10113175_

Round 1

Reviewer 1 Report

The authors of the review assessed inherited and acquired rhythm disturbances in sick syndrome, Brugada syndrome, and atrial fibrillation.

I have the following concerns:

  1. Could you please highlight the practical implications of the study?
  2. Could you please focus on possibilities of specific treatments variants of the rhythm disturbances, and possible personalized treatment?
  3. Could you please add some tables or figures in order to make the text more clear for the reader?

Author Response

We thank the Reviewer for the time dedicated to reading our manuscript and for the constructive comments reported.

In order to reply to the Reviewer’s concerns, we added one table per each rhythm disturbance. In particular, each table includes information on causes, modeling strategies, possible new insights on disease mechanism, as well as the translational impact that these developed models might have in terms of both clinical practical implications (diagnosis and prognosis), and the development of personalized medical treatments.

We hope the introduction of these comprehensive tables might satisfy the requests of the Reviewer and facilitate the reading of the manuscript.

Reviewer 2 Report

The paper is an excellent work on experimental and clinical findings on different rhythm disturbances. What is missing is subheading Brugada Syndrome in structural heart disease (arrhythmogenic cardiomyopathy).

Author Response

We thank the Reviewer for the time dedicated to reading our manuscript and appreciate her/his positive comments.

We acknowledge that Brugada syndrome displays characteristics typical of arrhythmogenic cardiomyopathy, as both are inherited diseases with an increased risk for arrhythmias and sudden death and hold overlapping features. However, there are important differences related to the structural defects associated with the two diseases. No structural defects can be appreciated in the hearts of Brugada syndrome patients at the angiographic or other imaging exams. Therefore, we prefer to add no further classification to this rhythm disturbance.

Reviewer 3 Report

This is a comprehensive review on major cardiac rhythm disturbances. Authors have to be congratulated for their meticulous work providing in detail information. My major concern about the present manuscript is that it such dense a writing format could potentially be difficult to follow. In that sense I would suggest the addition of Tables and Figures that would summarise and/or explain the main messages.

Author Response

We thank the Reviewer for the time dedicated and her/his constructive comments on our manuscript.

We are glad that this review manuscript provides a detailed overview of three rhythm disturbances that have strong clinical relevance. We acknowledge that as such, the manuscript is particularly lengthy and might render difficult its reading. We took advantage of the comment of the reviewers to add three tables, each per every considered rhythm disturbance. Each table includes information on the etiology when known, modeling strategies applied so far, and their implications for pathomechanism knowledge, as well as for diagnosis, prognosis, and therapy.

With this addition, we hope the reading of the manuscript can be eased.